# MVP: Multivariate polynomials for conditional data generation

## Abstract

Conditional Generative Adversarial Nets (cGANs) have been widely adopted for image generation. cGANs take i) a noise vector and ii) a conditional variable as input. The conditional variable can be discrete (e.g., a class label) or continuous (e.g., an input image) resulting into class-conditional (image) generation and image-to-image translation models, respectively. However, depending on whether the conditional variable is discrete or continuous, various cGANs employ substantially different deep architectures and loss functions for their training. In this paper, we propose a novel framework, called *MVP*, for conditional data generation. MVP resorts to multivariate polynomials of higher-order and treats in a unified way both discrete and continuous conditional variables. MVP is highly expressive, capturing higher-order auto- and cross-correlations of input variables (noise vector and conditional variable). Tailored sharing schemes are designed between the polynomial's parameter tensors, which result in simple recursive formulas. MVP can synthesize realistic images in both class-conditional and image-to-image translation tasks even in the absence of activation functions between the layers.

## 1 Introduction

Modelling high-dimensional distributions and generating samples from complex distributions are fundamental tasks in machine learning. Generative adversarial networks (GANs) (Goodfellow et al., 2014) have demonstrated spectacular results in the two tasks using both unsupervised (Miyato et al., 2018) and supervised (Brock et al., 2019) learning. In the unsupervised setting, (the generator of) a GAN accepts as input a noise vector $z_I$ and maps the noise vector to a high-dimensional output. The supervised models, called conditional Generative Adversarial Nets (cGANs) (Mirza & Osindero, 2014), accept both a noise vector $z_I$ and an additional conditional variable $z_{II}$ that facilitates the generation. The conditional variable can be discrete (e.g., a class or an attribute label) or continuous (e.g., a low-resolution image). The impressive results obtained with both discrete conditional input (Brock et al., 2019) and continuous conditional input (Park et al., 2019; Ledig et al., 2017) have led to a plethora of applications that range from text-to-image synthesis (Qiao et al., 2019) to deblurring (Yan & Wang, 2017) and medical analysis (You et al., 2019).

Despite the similarity in the formulation for discrete and continuous conditional input (i.e., learning the function $G(z_I, z_{II})$), the literature has focused on substantially different architectures and losses. Frequently, techniques are simultaneously developed, e.g., the self-attention in the class-conditional Self-Attention GAN (Zhang et al., 2019) and in the Attention-GAN (Chen et al., 2018) with continuous conditional input. This delays the progress since practitioners develop twice as many architectures and losses for every case. A couple of straightforward ideas can be employed to unify the behavior of the two conditional variable types. One idea is to use an encoder network to obtain representations that are independent of the conditional variable. This has two drawbacks: i) the network ignores the noise and a deterministic one-variable mapping is learned (Isola et al., 2017), ii) such encoder has not been successful so far for discrete conditional input. An alternative idea is to directly concatenate the labels in the latent space instead of finding an embedding. In AC-GAN (Odena et al., 2017) the class labels are concatenated with the noise; however, the model does not scale well beyond 10 classes. We argue that concatenation of the input is only capturing additive correlation and not higher-order interactions between the inputs. A detailed discussion is conducted on sec. D (in the Appendix).

A polynomial expansion with respect to the input variables can capture such higher-order correlations. Π-Net (Chrysos et al., 2020) casts the function approximation into a polynomial expansion of a single input variable. By concatenating the input variables, we can express the function approximation as a polynomial of the fused variable. However, the concatenation reduces the flexibility of the model significantly, e.g., it enforces the same order of expansion with respect to the different variables and it only allows the same parameter sharing scheme to all variables.

We introduce a multivariate framework, called *MVP*, for conditional data generation. MVP resorts to multivariate polynomials with two input variables, i.e., $z_{\mathrm{I}}$ for the noise vector and $z_{\mathrm{II}}$ for the conditional variable. MVP captures higher-order auto- and cross-correlations between the variables. By imposing a tailored structure in the higher-order interactions, we obtain an intuitive, recursive formulation for MVP. The formulation is flexible and enables different constraints to be applied to each variable and its associated parameters. The formulation can be trivially extended to $M$ input variables. In summary, our contributions are the following:

- We introduce a framework, called MVP, that expresses a high-order, multivariate polynomial for conditional data generation. Importantly, MVP treats both discrete and continuous conditional variables in a unified way.

- We offer an in-depth relationship with state-of-the-art works, such as SPADE (Park et al., 2019), that can be interpreted as polynomial expansions. We believe this perspective better explains the success of such architectures and offers a new direction for their extension.

- MVP is trained on *eight different datasets* for both class-conditional generation and image-to-image translation tasks. The trained models rely on both input variables, i.e., they do not ignore the noise vector.

- To illustrate the expressivity of the model, we also experiment with generators that do not use activation functions between the layers. We verify that MVP can synthesize realistic images even in the absence of activation functions between the layers.

The source code of MVP will be published upon the acceptance of the paper.

## 2 RELATED WORK

The literature on conditional data generation is vast; dedicated surveys per task (Agnese et al., 2019; Wu et al., 2017b) can be found for the interested reader. Below, we review representative works in conditional generation and then we summarize the recent progress in multiplicative interactions.

### 2.1 CONDITIONAL GENERATIVE MODELS

The challenging nature of image/video generation has led to a proliferation of conditional models. Although cGAN (Mirza & Osindero, 2014) is a general framework, since then the methods developed for conditional generation differ substantially depending on the type of conditional data. We present below representative works of the two categories, i.e., discrete and continuous conditional data, and their combination.

**Discrete conditional variable**: This is most frequently used for class-conditional generation (Miyato et al., 2018; Brock et al., 2019; Kaneko et al., 2019). Conditional normalization (Dumoulin et al., 2017; De Vries et al., 2017) techniques have been popular in the case of discrete conditional input, e.g., in generation of natural scenes images (Miyato et al., 2018; Brock et al., 2019). Conditional normalization cannot trivially generalize to a continuous conditional variable. In AC-GAN (Odena et al., 2017), they concatenate the class labels with the noise; however, their model does not scale well (i.e., they train one model per 10 classes). The aforementioned methods cannot be trivially used or modified for continuous conditional input. Text-to-image generation models (Qiao et al., 2019; Li et al., 2019; Zhang et al., 2018; Xu et al., 2018) use a specialized branch to embed the text labels.

**Continuous conditional variable**: The influential work of pix2pix (Isola et al., 2017) has become the reference point for continuous conditional input. The conditional input is embedded in a low-dimensional space (with an encoder), and then mapped to a high-dimensional output (through a decoder). The framework has been widely used for inverse tasks (Ledig et al., 2017; Pathak et al.,

2016; Wu et al., 2017a; Iizuka et al., 2017; Huang et al., 2017; Yu et al., 2018a; Grm et al., 2019; Xie et al., 2018; Yan & Wang, 2017), conditional pose generation (Ma et al., 2017; Siarohin et al., 2018; Liang et al., 2019), representation learning (Tran et al., 2017), conditional video generation (Wang et al., 2018a), generation from semantic labels (Wang et al., 2018b), image blending (Wu et al., 2019; Zhan et al., 2019). We recognize two major drawbacks in the aforementioned methods: a) they cannot be easily adapted for discrete conditional input, b) they learn a deterministic mapping, i.e., the noise is typically ignored. However, in many real applications, such as inverse tasks, the mapping is not one-to-one; there are multiple plausible outputs for every conditional input. The auxiliary losses used in such works, e.g., $\ell_1$ loss (Isola et al., 2017), perceptual loss (Ledig et al., 2017), are an additional drawback. Those losses both add hyper-parameters that require tuning and are domain-specific, thus it is challenging to transfer them to different domains or even different datasets. On the contrary, in our experiments, we do not use any additional loss.

**Discrete and continuous conditional variables**: Few works combine both discrete and continuous conditional inputs (Yu et al., 2018b; Xu et al., 2017; Lu et al., 2018). However, these methods include significant engineering (e.g., multiple discriminators (Xu et al., 2017), auxiliary losses), while often the generator learns to ignore the noise (similarly to the continuous conditional input). Antipov et al. (2017) design a generator for face aging. The generator combines continuous with discrete variables (age classes), however there is no Gaussian noise utilized, i.e., a deterministic transformation is learned for each input face. InfoGAN (Chen et al., 2016) includes both discrete and continuous conditional variables. However, the authors explicitly mention that additional losses are required, otherwise the generator is 'free to ignore' the additional variables.

The idea of Li et al. (2020) is most closely related to our work. They introduce a unifying framework for paired (Isola et al., 2017) and unpaired (Zhu et al., 2017a) learning. However, their framework assumes a continuous conditional input, while ours can handle discrete conditional input (e.g., class labels). In addition, their method requires a pre-trained teacher generator, while ours consists of a single generator trained end-to-end.

**Diverse data generation**: Conditional image generation often suffers from deterministic mappings, i.e., the noise variable has often negligible or negative impact in the generator (Zhu et al., 2017b; Isola et al., 2017). This has been tackled in the literature with additional loss terms and/or auxiliary network modules. A discussion of representative methods that tackle diverse generation is deferred to sec. I in the Appendix. In Table 1 the differences of the core techniques are summarized. Even though diverse generation is a significant task, we advocate that learning a generator does not ignore the input variables can be achieved without such additional loss terms. We highlight that diverse generation is a byproduct of MVP and not our main goal. Particularly, we believe that diverse images can be synthesized because the higher-order correlations of the input variables are captured effectively the proposed method.

Table 1: Comparison of techniques used for diverse, conditional generation. The majority of the methods insert additional loss terms, while some of them even require additional networks to be trained to achieve diverse generation results. MVP learns a non-deterministic mapping without additional networks or loss terms, thus simplifying the training. Nevertheless, as we empirically exhibit in sec. H.7, dedicated works that tackle diverse generation can be used in conjunction with the proposed MVP to further boost the diversity of the synthesized images.

| Methods for diverse generation. | | |
|---|---|---|
| Model | additional loss terms | auxiliary networks |
| BicycleGAN (Zhu et al., 2017b) | ✓ | ✓ |
| Yang et al. (2019); Lee et al. (2019) | ✓ | ✗ |
| Huang et al. (2018); Lee et al. (2020) | ✓ | ✓ |
| MVP (ours) | ✗ | ✗ |

## 2.2 MULTIPLICATIVE INTERACTIONS

Multiplicative connections have long been adopted in computer vision and machine learning (Shin & Ghosh, 1991; Hochreiter & Schmidhuber, 1997; Bahdanau et al., 2015). The idea is to combine the inputs through elementwise products or other diagonal forms. Even though multiplicative connections

have successfully been applied to different tasks, until recently there was no comprehensive study of their expressivity versus the standard feedforward networks. Jayakumar et al. (2020) include the proof that second order multiplicative operators can represent a greater class of functions than classic feed-forward networks. Even though we capitalize on the theoretical argument, our framework can express any higher-order interactions while the framework of Jayakumar et al. (2020) is limited to second order interactions.

Table 2: Comparison of attributes of polynomial-like neural networks. Even though the architectures of Karras et al. (2019); Chen et al. (2019); Park et al. (2019) were not posed as polynomial expansions, we believe that their success can be (partly) attributed to the polynomial expansion (please check sec. F for further information). Π-Net and StyleGAN are not designed for conditional data generation. In practice, learning complex distributions requires high-order polynomial expansions; this can be effectively achieved with products of polynomials as detailed in sec. 3.2. Only Π-Net and MVP include such a formulation. Additionally, the only work that enables multiple conditional variables (and includes experiments with both continuous and discrete conditional variables) is the proposed MVP.

| Attributes of polynomial-like networks. | | | | |
|---|---|---|---|---|
| Model | product of polynomials | discrete cond.variable | continuous cond. variable | multiple cond. variables |
| Π-Net (Chrysos et al., 2020) | ✓ | ✗ | ✗ | ✗ |
| StyleGAN (Karras et al., 2019) | ✗ | ✗ | ✗ | ✗ |
| sBN (Chen et al., 2019) | ✗ | ✓ | ✗ | ✗ |
| SPADE (Park et al., 2019) | ✗ | ✗ | ✓ | ✗ |
| MVP (ours) | ✓ | ✓ | ✓ | ✓ |

Higher-order interactions have been studied in the tensor-related literature (Kolda & Bader, 2009; Debals & De Lathauwer, 2017). However, their adaptation in modern deep architectures has been slower. Chrysos et al. (2020) propose high-order polynomial for mapping the input $z$ to the output $x = G(z)$. Π-Net focuses on a single input variable and cannot handle the multivariate cases that are the focus of this work. Three additional works that can be thought of as polynomial expansions are Karras et al. (2019); Park et al. (2019); Chen et al. (2019). The three works were originally introduced as (conditional) normalization variants, but we attribute their improvements in the expressiveness of their polynomial expansions. Under the polynomial expansion perspective, they can be expressed as special cases of the proposed MVP. A detailed discussion is conducted in sec. F in the Appendix. We believe that the proposed framework offers a direction to further extend the results of such works, e.g., by allowing more than one conditional variables.

## 3 METHOD

The framework for a multivariate polynomial with a two-variable input is introduced (sec. 3.1). The derivation, further intuition and additional models are deferred to the Appendix (sec. B). The crucial technical details, including the stability of the polynomial, are developed in sec. 3.2. We emphasize that a multivariate polynomial can approximate any function (Stone, 1948; Nikol'skii, 2013), i.e., a multivariate polynomial is a universal approximator.

**Notation**: Tensors/matrices/vectors are symbolized by calligraphic/uppercase/lowercase boldface letters e.g., $\mathcal{W}, \boldsymbol{W}, \boldsymbol{w}$. The *mode-$m$ vector product* of $\mathcal{W}$ (of order $M$) with a vector $\boldsymbol{u} \in \mathbb{R}^{I_m}$ is $\mathcal{W} \times_m \boldsymbol{u}$ and results in a tensor of order $M - 1$. We assume that $\prod_{i=a}^{b} x_i = 1$ when $a > b$. The core symbols are summarized in Table 3, while a detailed tensor notation is deferred to the Appendix (sec. B.1).

Table 3: Symbols

| Symbol | Role |
|---|---|
| $N$ | Expansion order of the polynomial |
| $k$ | Rank of the decompositions |
| $z_{\text{I}}, z_{\text{II}}$ | Inputs to the polynomial |
| $n, \rho$ | Auxiliary variables |
| $\mathcal{W}^{[n,\rho]}$ | Parameter tensor of the polynomial |
| $\boldsymbol{U}_{[n]}, \boldsymbol{C}, \boldsymbol{\beta}$ | Learnable parameters |
| $*$ | Hadamard product |

### 3.1 TWO INPUT VARIABLES

Given two input variables [1] $z_\mathrm{I}, z_\mathrm{II} \in \mathbb{K}^d$ where $\mathbb{K} \subseteq \mathbb{R}$ or $\mathbb{K} \subseteq \mathbb{N}$, the goal is to learn a function $G : \mathbb{K}^{d \times d} \to \mathbb{R}^o$ that captures the higher-degree interactions between the elements of the two inputs. We can learn such higher-degree interactions as polynomials of two input variables. A polynomial of expansion order $N \in \mathbb{N}$ with output $x \in \mathbb{R}^o$ has the form:

$$x = G(z_\mathrm{I}, z_\mathrm{II}) = \sum_{n=1}^{N} \sum_{\rho=1}^{n+1} \left( \mathcal{W}^{[n,\rho]} \prod_{j=2}^{\rho} \times_j z_\mathrm{I} \prod_{\tau=\rho+1}^{n+1} \times_\tau z_\mathrm{II} \right) + \beta \tag{1}$$

where $\beta \in \mathbb{R}^o$ and $\mathcal{W}^{[n,\rho]} \in \mathbb{R}^{o \times \prod_{m=1}^{n} \times_m d}$ for $n \in [1, N], \rho \in [1, n+1]$ are the learnable parameters. The expansion depends on two (independent) variables, hence we use the $n$ and $\rho$ as auxiliary variables. The two products of (1) do not overlap, i.e., the first multiplies the modes $[2, \rho]$ (of $\mathcal{W}^{[n,\rho]}$) with $z_\mathrm{I}$ and the other multiplies the modes $[\rho + 1, n+1]$ with $z_\mathrm{II}$.

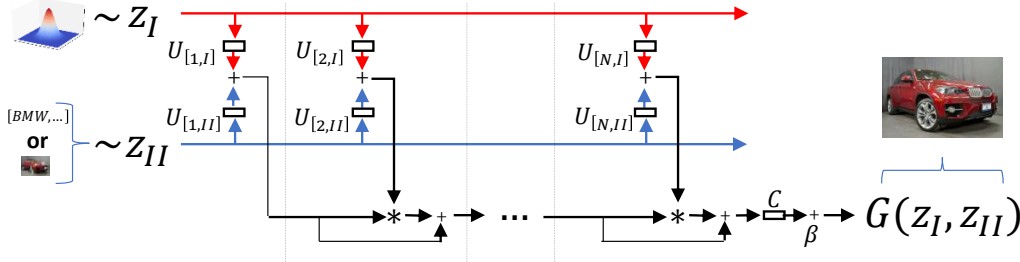

Figure 1: Abstract schematic for $N^{th}$ order approximation of $x = G(z_I, z_{II})$. The inputs $z_I, z_{II}$ are symmetric in our formulation. We denote with $z_I$ a sample from a prior distribution (e.g., Gaussian), while $z_{II}$ symbolizes a sample from a conditional input (e.g., class label or low-resolution image).

**Recursive relationship**: The aforementioned derivation can be generalized to an arbitrary expansion order. The recursive formula for an arbitrary order $N \in \mathbb{N}$ is the following:

$$x_n = x_{n-1} + \left( U_{[n,I]}^T z_\mathrm{I} + U_{[n,II]}^T z_\mathrm{II} \right) * x_{n-1} \tag{2}$$

for $n = 2, \ldots, N$ with $x_1 = U_{[1,I]}^T z_\mathrm{I} + U_{[1,II]}^T z_\mathrm{II}$ and $x = C x_N + \beta$. The parameters $C \in \mathbb{R}^{o \times k}, U_{[n,\phi]} \in \mathbb{R}^{d \times k}$ for $n = 1, \ldots, N$ and $\phi = \{I, II\}$ are learnable.

The intuition behind this model is the following: An embedding is initially found for each of the two input variables, then the two embeddings are added together and they are multiplied elementwise with the previous approximation. The different embeddings for each of the input variables allows us to implement $U_{[n,I]}$ and $U_{[n,II]}$ with different constraints, e.g., $U_{[n,I]}$ to be a dense layer and $U_{[n,II]}$ to be a convolution.

### 3.2 MODEL EXTENSIONS AND TECHNICAL DETAILS

There are three limitations in (2). Those are the following: a) (2) describes a polynomial expansion of a two-variable input, b) each expansion order requires additional layers, c) high-order polynomials might suffer from unbounded values. Those limitations are addressed below.

Our model can be readily extended beyond two-variable input; an extension with three-variable input is developed in sec. C. The pattern (for each order) is similar to the two-variable input: a) a different embedding is found for each input variable, b) the embeddings are added together, c) the result is multiplied elementwise with the representation of the previous order.

The polynomial expansion of (2) requires $\Theta(N)$ layers for an $N^{th}$ order expansion. That is, each new order $n$ of expansion requires new parameters $U_{[n,I]}$ and $U_{[n,II]}$. However, the order of expansion

---

[1]To avoid cluttering the notation we use same dimensionality for the two inputs. However, the derivations apply for different dimensionalities, only the dimensionality of the tensors change slightly.

can be increased without increasing the parameters substantially. To that end, we can capitalize on the product of polynomials. Specifically, let $N_1$ be the order of expansion of the first polynomial. The output of the first polynomial is fed into a second polynomial, which has expansion order of $N_2$. Then, the output of the second polynomial will have an expansion order of $N_1 \cdot N_2$. The product of polynomials can be used with arbitrary number of polynomials; it suffices the output of the $\tau^{th}$ polynomial to be the input to the $(\tau + 1)^{th}$ polynomial. For instance, if we assume a product of $\Phi \in \mathbb{N}$ polynomials, where each polynomial has an expansion order of two, then the polynomial expansion is of $2^\Phi$ order. In other words, we need $\Theta(\log_2(N))$ layers to achieve an $N^{th}$ order expansion.

In algebra, higher-order polynomials are unbounded and can thus suffer from instability for large values. To avoid such instability, we take the following three steps: a) MVP samples the noise vector from the uniform distribution, i.e., from the bounded interval of $[-1, 1]$, b) a hyperbolic tangent is used in the output of the generator as a normalization, i.e., it constrains the outputs in the bounded interval of $[-1, 1]$, c) batch normalization (Ioffe & Szegedy, 2015) is used to convert the representations to zero-mean. We emphasize that in GANs the hyperbolic tangent is the default activation function in the output of the generator, hence it is not an additional requirement of our method. Additionally, in our preliminary experiments, the uniform distribution can be changed for a Gaussian distribution without any instability. A theoretical analysis on the bounds of such multivariate polynomials would be an interesting subject for future work.

## 4 EXPERIMENTS

The proposed MVP is empirically evaluated in three settings: a) a class-conditional generation, i.e., with discrete conditional input, b) an image-to-image translation, i.e., with continuous conditional input, c) a mixed conditional setting with two conditional variables. The goal is to showcase how MVP can be used with both discrete and continuous conditional inputs. Even though architectures specialized for a single task (e.g., Ledig et al. (2017)) perform well in that task, their well-selected inductive biases (e.g., perceptual or $\ell_1$ loss) do not generalize well in other domains or different conditional inputs. Hence, our goal is not to demonstrate state-of-the-art results in specific tasks, but rather to propose one generic formulation. Further experiments (e.g., class-conditional generation with SVHN or MNIST to SVHN translation; sec H), the details on the datasets and the evaluation metrics (sec. G) are deferred to the Appendix. Throughout the experimental section, we reserve the symbol $z_{\mathrm{II}}$ for the conditional input (e.g., a class label).

Our framework, e.g., (2), does not include any activation functions. To verify the expressivity of our framework, we maintain the same setting for the majority of the experiments below. Particularly, the generator does not have activation functions between the layers; there is only a hyperbolic tangent in the output space for normalization. Training a generator without activation functions between the layers also emerged in $\Pi$-Net (Chrysos et al., 2020), where the authors demonstrate the challenges in such framework. However, we conduct one experiment using a strong baseline with activation functions. That is, a comparison with SNGAN (Miyato & Koyama, 2018) in class-conditional generation is performed (sec. 4.1).

**Baselines:** '$\Pi$-Net-SICONC' implements a polynomial expansion of a single variable, i.e., by concatenating all the input variables. 'SPADE' implements a polynomial expansion with respect to the conditional variable. Also, 'GAN-CONC' and 'GAN-ADD' are added as baselines, where we replace the Hadamard products with concatenation and addition respectively. An abstract schematic of the differences between the compared polynomial methods is depicted in Fig. 6, while a detailed description of all methods is deferred to sec. G. Each experiment is conducted **five** times and the mean and the standard deviation are reported.

### 4.1 CLASS-CONDITIONAL GENERATION

The first experiment is on class-conditional generation, where the conditional input is a class label in the form of one-hot vector. Two types of networks are utilized: a) a resnet-based generator (SNGAN), b) a polynomial generator ($\Pi$-Net) based on Chrysos et al. (2020). The former network has exhibited strong performance the last few years, while the latter bears resemblance to the formulation we propose in this work.

Table 4: Quantitative evaluation on class-conditional generation with resnet-based generator (i.e., SNGAN). Higher Inception Score (IS) (Salimans et al., 2016) (lower Frechet Inception Distance (FID) (Heusel et al., 2017)) indicates better performance. The baselines improve the IS of SNGAN, however they cannot improve the FID. Nevertheless, SNGAN-MVP improves upon all the baselines in both the IS and the FID.

| class-conditional generation on CIFAR10 | | |
| --- | --- | --- |
| Model | IS ($\uparrow$) | FID ($\downarrow$) |
| SNGAN | $8.30 \pm 0.11$ | $14.70 \pm 0.97$ |
| SNGAN-CONC | $8.50 \pm 0.49$ | $30.65 \pm 3.55$ |
| SNGAN-ADD | $8.65 \pm 0.11$ | $15.47 \pm 0.74$ |
| SNGAN-SPADE | $8.69 \pm 0.19$ | $21.74 \pm 0.73$ |
| SNGAN-MVP | $\mathbf{8.77 \pm 0.12}$ | $\mathbf{14.22 \pm 0.66}$ |

**Resnet-based generator:** The experiment is conducted by augmenting the resnet-based generator of SNGAN. The quantitative results are in Table 4 and synthesized samples are illustrated in Fig. 2(a). SNGAN-MVP improves upon all the baselines in both the Inception score (IS) (Salimans et al., 2016) and the FID (Heusel et al., 2017). The proposed formulation enables inter-class interpolations. That is, the noise $z_{\mathrm{I}}$ is fixed, while the class $z_{\mathrm{II}}$ is interpolated. In Fig. 2(b) and Fig. 2(c), intra-class and inter-class linear interpolations are illustrated respectively. Both the quantitative and the qualitative results exhibit the effectiveness of our framework.

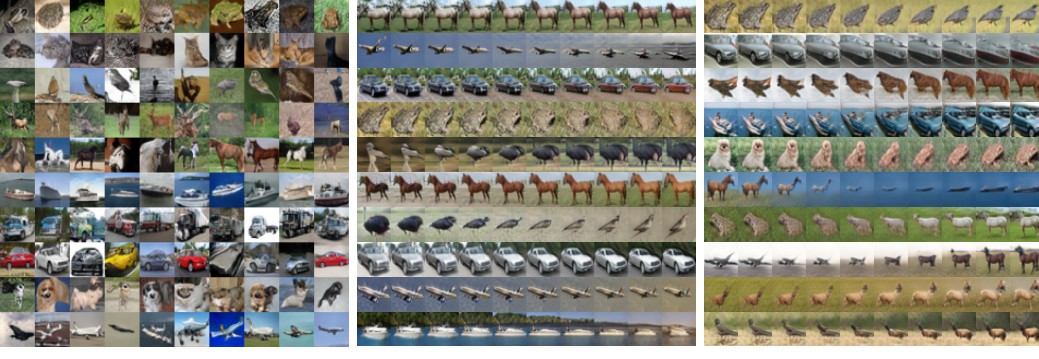

(a) Random samples per class          (b) Intra-class interpolation          (c) Inter-class interpolation

Figure 2: Synthesized images by MVP in the class-conditional CIFAR10 (with resnet-based generator): (a) Random samples where each row depicts the same class, (b) Intra-class linear interpolation from a source to the target, (c) inter-class linear interpolation. In inter-class interpolation, the class labels of the leftmost and rightmost images are one-hot vectors, while the rest are interpolated in-between; the resulting images are visualized. In all three cases, MVP synthesizes realistic images.

Table 5: Quantitative evaluation on class-conditional generation with Π-Net-based generator. In CIFAR10, there is a considerable improvement on the IS, while in Cars196 FID drops dramatically with MVP. We hypothesize that the dramatic improvement in Cars196 arises because of the correlations of the classes. For instance, the SUV cars (of different carmakers) share several patterns, which are captured by our high-order interactions, while they might be missed when learning different normalization statistics per class.

| class-conditional generation on CIFAR10 | | | class-conditional generation on Cars196 | |
| --- | --- | --- | --- | --- |
| Model | IS ($\uparrow$) | FID ($\downarrow$) | Model | FID ($\downarrow$) |
| GAN-CONC | $3.73 \pm 0.32$ | $294.33 \pm 8.16$ | GAN-CONC | $240.45 \pm 16.79$ |
| GAN-ADD | $3.74 \pm 0.60$ | $298.53 \pm 16.54$ | GAN-ADD | $208.72 \pm 12.65$ |
| SPADE | $4.00 \pm 0.53$ | $294.21 \pm 16.33$ | SPADE | $168.19 \pm 39.71$ |
| Π-Net-SICONC | $6.65 \pm 0.60$ | $71.81 \pm 33.00$ | Π-Net-SICONC | $153.39 \pm 27.93$ |
| Π-Net | $7.54 \pm 0.16$ | $37.26 \pm 1.86$ | Π-Net | $120.40 \pm 28.65$ |
| MVP | $\mathbf{7.87 \pm 0.21}$ | $\mathbf{34.35 \pm 2.68}$ | MVP | $\mathbf{55.48 \pm 3.16}$ |

$\Pi$**-Net-based generator:**    A product of polynomials, based on $\Pi$-Net, is selected as the baseline architecture for the generator. $\Pi$-Net has conditional batch normalization (CBN) in the generator, while in the rest compared methods CBN is replaced by batch normalization. The results in CIFAR10 are summarized in Table 5 (left), where MVP outperforms all the baselines by a large margin. An additional experiment is performed in Cars196 that has 196 classes. The results in Table 5 (right) depict a substantial improvement over the all the baselines (53.9% reduction over the best-performing baseline). We should note that the baseline was not built for conditional generation, however we have done our best effort to optimize the respective hyper-parameters. We hypothesize that the improvement arises because of the correlations of the classes. That is, the 196 classes might be correlated (e.g., the SUV cars of different carmakers share several patterns). Such correlations are captured by our framework, while they might be missed when learning different normalization statistics per class. Overall, MVP synthesizes plausible images (Fig. 11) even in the absence of activation functions.

## 4.2    Continuous conditional input

The performance of MVP is scrutinized in tasks with continuous conditional input, e.g., super-resolution. The conditional input $z_\Pi$ is an input image, e.g., a low-resolution sample or a corrupted sample. Even though the core architecture remains the same, a single change is made in the structure of the discriminator: Motivated by (Miyato & Koyama, 2018), we include an elementwise product of $z_\Pi$ with the real/fake image in the discriminator. This stabilizes the training and improves the results. A wealth of literature is available on such continuous conditional inputs (sec. 2.1), however we select the challenging setting of using a generator without activation functions between the layers.

Table 6: Quantitative evaluation on super-resolution with $\Pi$-Net-based generator on Cars196. The task on the left is super-resolution $16\times$, while on the right the task is super-resolution $8\times$. Our variant of SPADE, i.e., SPADE-MVP (details in sec. G), vastly improves the original SPADE. The full two-variable model, i.e., MVP, outperforms the compared methods.

| Super-resolution $16\times$ Cars196 | | Super-resolution $8\times$ Cars196 | |
| --- | --- | --- | --- |
| Model | FID ($\downarrow$) | Model | FID ($\downarrow$) |
| SPADE | $111.75 \pm 13.41$ | SPADE | $119.18 \pm 14.82$ |
| $\Pi$-Net-SICONC | $80.16 \pm 12.42$ | $\Pi$-Net-SICONC | $186.42 \pm 40.84$ |
| SPADE-MVP | $72.63 \pm 3.18$ | SPADE-MVP | $64.76 \pm 8.26$ |
| MVP | $\mathbf{60.42 \pm 6.19}$ | MVP | $\mathbf{62.76 \pm 4.37}$ |

The experiments are performed in (a) super-resolution, (b) block-inpainting. Super-resolution assumes a low-resolution image is available, while in block inpainting, a (rectangular) part of the image is missing. The two tasks belong in the broader category of 'inverse tasks', and they are significant both for academic reasons but also for commercial reasons (Sood et al., 2018; You et al., 2019). Such inverse tasks are underdetermined; each input image corresponds to several plausible output images.

The FID scores in Cars196 for the task of super-resolution are reported in Table 6. In super-resolution $16\times$, $z_\Pi$ has 48 dimensions, while in super-resolution $8\times$, $z_\Pi$ has 192 dimensions. Notice that the performance of $\Pi$-Net-SICONC deteriorates substantially when the dimensionality of the conditional variable increases. That validates our intuition about the concatenation in the input of the generator (sec. E). We also report the SPADE-MVP, which captures higher-order correlations with respect to the first variable as well (further details in sec. G). The proposed SPADE-MVP outperforms the original SPADE, however it cannot outperform the full two-variable model, i.e., MVP. MVP maintains outperforms all baselines by a large margin.

The qualitative results on (a) super-resolution $8\times$ on CelebA, (b) super-resolution $8\times$ on Cars196, (c) super-resolution $16\times$ on Cars196 are illustrated in Fig. 3. Similarly the qualitative results on block-inpainting are visualized in Fig. 11. For each conditional image, different noise vectors $z_I$ are sampled. Notice that the corresponding synthesized images differ in the fine-details. For instance, changes in the mouth region, the car type or position and even background changes are observed. Thus, MVP results in high-resolution images that i) correspond to the conditional input, ii) vary in fine-details. Similar variation has emerged even when the source and the target domains differ substantially, e.g., in the translation of MNIST digits to SVHN digits (sec. H.3). We should mention that regularization techniques have been proposed specifically for image-to-image translation, e.g.,

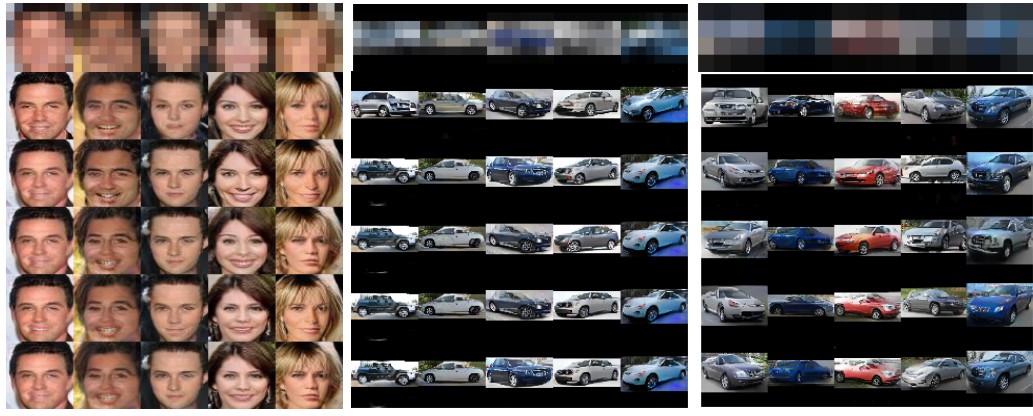

| (a) Super-resolution 8× | (b) Super-resolution 8× | (c) Super-resolution 16× |

Figure 3: Synthesized images for super-resolution by (a), (b) 8×, (c) 16×. The first row depicts the conditional input (i.e., low-resolution image). The rows 2-6 depict outputs of the MVP when a noise vector is sampled per row. Notice how the noise changes (a) the smile or the pose of the head, (b) the color, car type or even the background, (c) the position of the car.

Yang et al. (2019); Lee et al. (2019). However, such works utilize additional losses and even require additional networks for training, which makes the training more computationally heavy and more sensitive to design choices.

## 5 CONCLUSION

The topic of conditional data generation is the focus of this work. A multivariate polynomial model, called MVP, is introduced. MVP approximates a function $G(z_I, z_{II})$ with inputs $z_I$ (e.g., sample from a Gaussian distribution) and $z_{II}$ (e.g., class or low-resolution image). MVP resorts to multivariate polynomials with arbitrary conditional inputs, which capture high-order correlations of the inputs. The empirical evaluation confirms that our framework can synthesize realistic images in both class-conditional generation (trained on CIFAR10, Cars196 and SVHN), attribute-guided generation and image-to-image translation (i.e., super-resolution, block-inpainting, edges-to-shoes, edges-to-handbag, MNIST-to-SVHN). We also showcase that it can be extended to three-variable input with class-conditional super-resolution. In addition to conditional data generation, the proposed framework can be used in tasks requiring fusion of different types of variables.

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

## A    Summary of sections in the Appendix

In the following sections, further details and derivations are provided to elaborate the details of the MVP. Specifically, in sec. B the decomposition and related details on the method are developed. The extension of our method beyond two-input variables is studied in sec. C. A method frequently used in the literature for fusing information is concatenation; we analyze how concatenation captures only additive and not more complex correlations (e.g., multiplicative) in sec. D. The differences from Π-Net (Chrysos et al., 2020) is explored in sec. E. In sec. F, some recent (conditional) data generation methods are cast into the polynomial neural network framework and their differences from the proposed framework are analyzed. The experimental details including the evaluation metrics and details on the baselines are developed in sec. G. In sec. H, additional experimental results are included. Lastly, the differences from works that perform diverse generation are explored in sec. I.

## B    Method derivations

In this section, we expand on the method details, including the scalar output case or the notation. Specifically, a more detailed notation is determined in sec. B.1; the scalar output case is analyzed in sec. B.2. In sec. B.3 a second order expansion is assumed to illustrate the connection between the polynomial expansion and the recursive formula. Sequentially, we derive an *alternative* model with different factor sharing. This model, called Nested-MVP, has a nested factor sharing format (sec. B.4).

### B.1    Notation

Our derivations rely on tensors (i.e., multidimensional equivalent of matrices) and (tensor) products. We relay below the core notation used in our work, the interested reader can find further information in the tensor-related literature (Kolda & Bader, 2009; Debals & De Lathauwer, 2017).

**Symbols of variables**: Tensors/matrices/vectors are symbolized by calligraphic/uppercase/lowercase boldface letters e.g., $\mathcal{W}, \boldsymbol{W}, \boldsymbol{w}$.

**Matrix products**: The *Hadamard* product of $\boldsymbol{A}, \boldsymbol{B} \in \mathbb{R}^{I \times N}$ is defined as $\boldsymbol{A} * \boldsymbol{B}$ and is equal to $a_{(i,j)} b_{(i,j)}$ for the $(i,j)$ element. The *Khatri-Rao* product of matrices $\boldsymbol{A} \in \mathbb{R}^{I \times N}$ and $\boldsymbol{B} \in \mathbb{R}^{J \times N}$ is

denoted by $\boldsymbol{A} \odot \boldsymbol{B}$ and yields a matrix of dimensions $(IJ) \times N$. The Khatri-Rao product for a set of matrices $\{\boldsymbol{A}_{[m]} \in \mathbb{R}^{I_m \times N}\}_{m=1}^{M}$ is abbreviated by $\boldsymbol{A}_{[1]} \odot \boldsymbol{A}_{[2]} \odot \cdots \odot \boldsymbol{A}_{[M]} \doteq \bigodot_{m=1}^{M} \boldsymbol{A}_{[m]}$.

**Tensors**: Each element of an $M^{th}$ order tensor $\mathcal{W}$ is addressed by $M$ indices, i.e., $(\mathcal{W})_{i_1,i_2,\dots,i_M} \doteq w_{i_1,i_2,\dots,i_M}$. An $M^{th}$-order tensor $\mathcal{W}$ is defined over the tensor space $\mathbb{R}^{I_1 \times I_2 \times \cdots \times I_M}$, where $I_m \in \mathbb{Z}$ for $m = 1, 2, \dots, M$. The *mode-m unfolding* of a tensor $\mathcal{W} \in \mathbb{R}^{I_1 \times I_2 \times \cdots \times I_M}$ maps $\mathcal{W}$ to a matrix $\boldsymbol{W}_{(m)} \in \mathbb{R}^{I_m \times \bar{I}_m}$ with $\bar{I}_m = \prod_{\substack{k=1 \\ k \neq m}}^{M} I_k$ such that the tensor element $w_{i_1,i_2,\dots,i_M}$ is mapped to the matrix element $w_{i_m,j}$ where $j = 1 + \sum_{\substack{k=1 \\ k \neq m}}^{M} (i_k - 1)J_k$ with $J_k = \prod_{\substack{n=1 \\ n \neq m}}^{k-1} I_n$. The *mode-m vector product* of $\mathcal{W}$ with a vector $\boldsymbol{u} \in \mathbb{R}^{I_m}$, denoted by $\mathcal{W} \times_m \boldsymbol{u} \in \mathbb{R}^{I_1 \times I_2 \times \cdots \times I_{m-1} \times I_{m+1} \times \cdots \times I_M}$, results in a tensor of order $M - 1$:

$$(\mathcal{W} \times_m \boldsymbol{u})_{i_1,\dots,i_{m-1},i_{m+1},\dots,i_M} = \sum_{i_m=1}^{I_m} w_{i_1,i_2,\dots,i_M} u_{i_m}. \tag{3}$$

We denote $\mathcal{W} \times_1 \boldsymbol{u}^{(1)} \times_2 \boldsymbol{u}^{(2)} \times_3 \cdots \times_M \boldsymbol{u}^{(M)} \doteq \mathcal{W} \prod_{m=1}^{m} \times_m \boldsymbol{u}^{(m)}$.

The *CP decomposition* (Kolda & Bader, 2009) factorizes a tensor into a sum of component rank-one tensors. The rank-$R$ CP decomposition of an $M^{th}$-order tensor $\mathcal{W}$ is written as:

$$\mathcal{W} \doteq [\![\boldsymbol{U}_{[1]}, \boldsymbol{U}_{[2]}, \dots, \boldsymbol{U}_{[M]}]\!] = \sum_{r=1}^{R} \boldsymbol{u}_r^{(1)} \circ \boldsymbol{u}_r^{(2)} \circ \cdots \circ \boldsymbol{u}_r^{(M)}, \tag{4}$$

where $\circ$ is the vector outer product. The factor matrices $\{\boldsymbol{U}_{[m]} = [\boldsymbol{u}_1^{(m)}, \boldsymbol{u}_2^{(m)}, \cdots, \boldsymbol{u}_R^{(m)}] \in \mathbb{R}^{I_m \times R}\}_{m=1}^{M}$ collect the vectors from the rank-one components. By considering the mode-1 unfolding of $\mathcal{W}$, the CP decomposition can be written in matrix form as:

$$\boldsymbol{W}_{(1)} \doteq \boldsymbol{U}_{[1]} \left( \bigodot_{m=M}^{2} \boldsymbol{U}_{[m]} \right)^T \tag{5}$$

The following lemma is useful in our method:

**Lemma 1.** *For a set of $N$ matrices $\{\boldsymbol{A}_{[\nu]} \in \mathbb{R}^{I_\nu \times K}\}_{\nu=1}^{N}$ and $\{\boldsymbol{B}_{[\nu]} \in \mathbb{R}^{I_\nu \times L}\}_{\nu=1}^{N}$, the following equality holds:*

$$(\bigodot_{\nu=1}^{N} \boldsymbol{A}_{[\nu]})^T \cdot (\bigodot_{\nu=1}^{N} \boldsymbol{B}_{[\nu]}) = (\boldsymbol{A}_{[1]}^T \cdot \boldsymbol{B}_{[1]}) * \dots * (\boldsymbol{A}_{[N]}^T \cdot \boldsymbol{B}_{[N]}) \tag{6}$$

An indicative proof can be found in the Appendix of Chrysos et al. (2019).

## B.2 SCALAR OUTPUT

The proposed formulation expresses higher-order interactions of the input variables. To elaborate that, we develop the single output case below. That is, we focus on an element $\tau$ of the output vector, e.g., a single pixel. In the next few paragraphs, we consider the case of a scalar output $x_\tau$, with $\tau \in [1, o]$ when the input variables are $\boldsymbol{z}_{\text{I}}, \boldsymbol{z}_{\text{II}} \in \mathbb{K}^d$. To avoid cluttering the notation we only refer to the scalar output with $x_\tau$ in the next few paragraphs.

As a reminder, the polynomial of expansion order $N \in \mathbb{N}$ with output $\boldsymbol{x} \in \mathbb{R}^o$ has the form:

$$\boldsymbol{x} = G(\boldsymbol{z}_{\text{I}}, \boldsymbol{z}_{\text{II}}) = \sum_{n=1}^{N} \sum_{\rho=1}^{n+1} \left( \mathcal{W}^{[n,\rho]} \prod_{j=2}^{\rho} \times_j \boldsymbol{z}_{\text{I}} \prod_{\tau=\rho+1}^{n+1} \times_\tau \boldsymbol{z}_{\text{II}} \right) + \boldsymbol{\beta} \tag{7}$$

We assume a second order expansion ($N = 2$) and let $\tau$ denote an arbitrary scalar output of $\boldsymbol{x}$. The first order correlations can be expressed through the sums $\sum_{\lambda=1}^{d} w_{\tau,\lambda}^{[1,1]} z_{\text{II},\lambda}$ and $\sum_{\lambda=1}^{d} w_{\tau,\lambda}^{[1,2]} z_{\text{I},\lambda}$. The second order correlations include both auto- and cross-correlations. The tensors $\mathcal{W}^{[2,1]}$ and $\mathcal{W}^{[2,3]}$ capture the auto-correlations, while the tensor $\mathcal{W}^{[2,2]}$ captures the cross-correlations.

$$x_\tau = \left[ w^{[1,1]}_{\tau,1}, w^{[1,1]}_{\tau,2}, \cdots, w^{[1,1]}_{\tau,d} \right] \begin{bmatrix} z_{II,1} \\ \vdots \\ z_{II,d} \end{bmatrix} + \left[ w^{[1,2]}_{\tau,1}, w^{[1,2]}_{\tau,2}, \cdots, w^{[1,2]}_{\tau,d} \right] \begin{bmatrix} z_{I,1} \\ \vdots \\ z_{I,d} \end{bmatrix} + \left[ \ \cdots \ z_{II,\lambda} \ \cdots \ \right] \left[ w^{[2,1]}_{\tau,\lambda,\mu} \right] \begin{bmatrix} \vdots \\ z_{II,\mu} \\ \vdots \end{bmatrix} +$$

$$\left[ \ \cdots \ z_{I,\lambda} \ \cdots \ \right] \left[ w^{[2,3]}_{\tau,\lambda,\mu} \right] \begin{bmatrix} \vdots \\ z_{I,\mu} \\ \vdots \end{bmatrix} + \left[ \ \cdots \ z_{I,\lambda} \ \cdots \ \right] \left[ w^{[2,2]}_{\tau,\lambda,\mu} \right] \begin{bmatrix} \vdots \\ z_{II,\mu} \\ \vdots \end{bmatrix}$$

Figure 4: Schematic for second order expansion with scalar output $x_\tau \in \mathbb{R}$. The abbreviations $z_{I,\lambda}, z_{I,\mu}$ are elements of $z_I$ with $\lambda, \mu \in [1, d]$. Similarly, $z_{II,\lambda}, z_{II,\mu}$ are elements of $z_{II}$. The first two terms (on the right side of the equation) are the first-order correlations; the next two terms are the second order auto-correlations. The last term expresses the second order cross-correlations.

A pictorial representation of the correlations are captured in Fig. 4. Collecting all the terms in an equation, each output is expressed as:

$$x_\tau = \beta_\tau + \sum_{\lambda=1}^d \left[ w^{[1,1]}_{\tau,\lambda} z_{II,\lambda} + w^{[1,2]}_{\tau,\lambda} z_{I,\lambda} + \sum_{\mu=1}^d w^{[2,1]}_{\tau,\lambda,\mu} z_{II,\lambda} z_{II,\mu} + \sum_{\mu=1}^d w^{[2,3]}_{\tau,\lambda,\mu} z_{I,\lambda} z_{I,\mu} + \sum_{\mu=1}^d w^{[2,2]}_{\tau,\lambda,\mu} z_{I,\lambda} z_{II,\mu} \right]$$
(8)

where $\beta_\tau \in \mathbb{R}$. Notice that all the correlations of up to second order are captured in equation 8.

### B.3 SECOND ORDER DERIVATION FOR TWO-VARIABLE INPUT

In all our derivations, the variables associated with the first input $z_I$ have an $I$ notation, e.g., $U_{[1,I]}$. Respectively for the second input $z_{II}$, the notation $II$ is used.

Even though equation 7 enables any order of expansion, the learnable parameters increase exponentially, therefore we can use a coupled factorization to reduce the parameters. Next, we derive the factorization for a second order expansion (i.e., $N = 2$) and then provide the recursive relationship that generalizes it for an arbitrary order.

**Second order derivation**: For a second order expansion (i.e., $N = 2$ in equation 1), we factorize each parameter tensor $\mathcal{W}^{[n,\rho]}$. We assume a coupled CP decomposition for each parameter as follows:

- Let $W^{[1,1]}_{(1)} = CU^T_{[1,II]}$ and $W^{[1,2]}_{(1)} = CU^T_{[1,I]}$ be the parameters for $n = 1$.
- Let $W^{[2,1]}_{(1)} = C(U_{[2,II]} \odot U_{[1,II]})^T$ and $W^{[2,3]}_{(1)} = C(U_{[2,I]} \odot U_{[1,I]})^T$ capture the second order correlations of a single variable ($z_{II}$ and $z_I$ respectively).
- The cross-terms are expressed in $\mathcal{W}^{[2,2]} \times_2 z_I \times_3 z_{II}$. The output of the $\tau$ element[2] is $\sum_{\lambda,\mu=1}^d w^{[2,2]}_{\tau,\lambda,\mu} z_{I,\lambda} z_{II,\mu}$. The product $\hat{\mathcal{W}}^{[2,2]} \times_2 z_{II} \times_3 z_I$ also results in the same elementwise expression. Hence, to allow for symmetric expression, we factorize the term $W^{[2,2]}_{(1)}$ as the sum of the two terms $C(U_{[2,II]} \odot U_{[1,I]})^T$ and $C(U_{[2,I]} \odot U_{[1,II]})^T$. For each of the two terms, we assume that the vector-valued inputs are accordingly multiplied.

The parameters $C \in \mathbb{R}^{o \times k}, U_{[m,\phi]} \in \mathbb{R}^{d \times k}$ ($m = 1, 2$ and $\phi = \{I, II\}$) are learnable. The aforementioned factorization results in the following equation:

$$x = CU^T_{[1,II]} z_{II} + CU^T_{[1,I]} z_I + C\left(U_{[2,II]} \odot U_{[1,II]}\right)^T \left(z_{II} \odot z_{II}\right) + C\left(U_{[2,I]} \odot U_{[1,I]}\right)^T \left(z_I \odot z_I\right) +$$
$$C\left(U_{[2,I]} \odot U_{[1,II]}\right)^T \left(z_I \odot z_{II}\right) + C\left(U_{[2,II]} \odot U_{[1,I]}\right)^T \left(z_{II} \odot z_I\right) + \beta$$
(9)

---

[2]An elementwise analysis (with a scalar output) is provided on the Appendix (sec. B.2).

This expansion captures the correlations (up to second order) of the two input variables $z_\mathrm{I}$, $z_\mathrm{II}$.

To make the proof more complete, we remind the reader that the recursive relationship (i.e., (2) in the main paper) is:

$$\boldsymbol{x}_n = \boldsymbol{x}_{n-1} + \left( \boldsymbol{U}_{[n,I]}^T \boldsymbol{z}_\mathrm{I} + \boldsymbol{U}_{[n,II]}^T \boldsymbol{z}_\mathrm{II} \right) * \boldsymbol{x}_{n-1} \tag{10}$$

for $n = 2, \ldots, N$ with $\boldsymbol{x}_1 = \boldsymbol{U}_{[1,I]}^T \boldsymbol{z}_\mathrm{I} + \boldsymbol{U}_{[1,II]}^T \boldsymbol{z}_\mathrm{II}$ and $\boldsymbol{x} = \boldsymbol{C}\boldsymbol{x}_N + \boldsymbol{\beta}$.

**Claim 1.** *The equation (9) is a special format of a polynomial that is visualized as in Fig. 1 of the main paper. Equivalently, prove that (9) follows the recursive relationship of (10).*

*Proof.* We observe that the first two terms of equation 9 are equal to $\boldsymbol{C}\boldsymbol{x}_1$ (from equation 10). By applying Lemma 1 in the terms that have Khatri-Rao product, we obtain:

$$\boldsymbol{x} = \boldsymbol{\beta} + \boldsymbol{C}\boldsymbol{x}_1 + \boldsymbol{C}\Big\{ \left( \boldsymbol{U}_{[2,II]}^T \boldsymbol{z}_\mathrm{II} \right) * \left( \boldsymbol{U}_{[1,II]}^T \boldsymbol{z}_\mathrm{II} \right) + \left( \boldsymbol{U}_{[2,I]}^T \boldsymbol{z}_\mathrm{I} \right) * \left( \boldsymbol{U}_{[1,I]}^T \boldsymbol{z}_\mathrm{I} \right) +$$

$$\left( \boldsymbol{U}_{[2,I]}^T \boldsymbol{z}_\mathrm{I} \right) * \left( \boldsymbol{U}_{[1,II]}^T \boldsymbol{z}_\mathrm{II} \right) + \left( \boldsymbol{U}_{[2,II]}^T \boldsymbol{z}_\mathrm{II} \right) * \left( \boldsymbol{U}_{[1,I]}^T \boldsymbol{z}_\mathrm{I} \right) \Big\} = \tag{11}$$

$$\boldsymbol{\beta} + \boldsymbol{C}\boldsymbol{x}_1 + \boldsymbol{C}\Big\{ \left[ \left( \boldsymbol{U}_{[2,I]}^T \boldsymbol{z}_\mathrm{I} \right) + \left( \boldsymbol{U}_{[2,II]}^T \boldsymbol{z}_\mathrm{II} \right) \right] * \boldsymbol{x}_1 \Big\} = \boldsymbol{C}\boldsymbol{x}_2 + \boldsymbol{\beta}$$

The last equation is precisely the one that arises from the recursive relationship from equation 10.

$\square$

To prove the recursive formula for the $N^{th}$ order expansion, a similar pattern as in sec.C of Poly-GAN (Chrysos et al., 2019) can be followed. Specifically, the difference here is that because of the two input variables, the auto- and cross-correlation variables should be included. Other than that, the same factor sharing is followed.

### B.4 NESTED-MVP MODEL FOR TWO-VARIABLE INPUT

The model proposed above (i.e., equation 10), relies on a single coupled CP decomposition, however a more flexible model can factorize each level with a CP decomposition. To effectively do that, we utilize learnable hyper-parameters $\boldsymbol{b}_{[n]} \in \mathbb{R}^\omega$ for $n \in [1, N]$, which act as scaling factors for each parameter tensor. Then, a polynomial of expansion order $N \in \mathbb{N}$ with output $\boldsymbol{x} \in \mathbb{R}^o$ has the form:

$$\boldsymbol{x} = G(\boldsymbol{z}_\mathrm{I}, \boldsymbol{z}_\mathrm{II}) = \sum_{n=1}^{N} \sum_{\rho=2}^{n+2} \left( \boldsymbol{\mathcal{W}}^{[n,\rho-1]} \times_2 \boldsymbol{b}_{[N+1-n]} \prod_{j=3}^{\rho} \times_j \boldsymbol{z}_\mathrm{I} \prod_{\tau=\rho+1}^{n+2} \times_\tau \boldsymbol{z}_\mathrm{II} \right) + \boldsymbol{\beta} \tag{12}$$

To demonstrate the factorization without cluttering the notation, we assume a second order expansion in equation 12.

**Second order derivation**: The second order expansion, i.e., $N = 2$, is derived below. We jointy factorize all parameters of equation 12 with a nested decomposition as follows:

- First order parameters : $\boldsymbol{W}_{(1)}^{[1,1]} = \boldsymbol{C}(\boldsymbol{A}_{[2,II]} \odot \boldsymbol{B}_{[2]})^T$ and $\boldsymbol{W}_{(1)}^{[1,2]} = \boldsymbol{C}(\boldsymbol{A}_{[2,I]} \odot \boldsymbol{B}_{[2]})^T$.

- Let $\boldsymbol{W}_{(1)}^{[2,1]} = \boldsymbol{C}\Big\{ \boldsymbol{A}_{[2,II]} \odot \Big[ \left( \boldsymbol{A}_{[1,II]} \odot \boldsymbol{B}_{[1]} \right) \boldsymbol{V}_{[2]} \Big] \Big\}^T$ and $\boldsymbol{W}_{(1)}^{[2,3]} = \boldsymbol{C}\Big\{ \boldsymbol{A}_{[2,I]} \odot \Big[ \left( \boldsymbol{A}_{[1,I]} \odot \boldsymbol{B}_{[1]} \right) \boldsymbol{V}_{[2]} \Big] \Big\}^T$ capture the second order correlations of a single variable ($\boldsymbol{z}_\mathrm{II}$ and $\boldsymbol{z}_\mathrm{I}$ respectively).

- The cross-terms are included in $\boldsymbol{\mathcal{W}}^{[2,2]} \times_2 \boldsymbol{b}_{[1]} \times_3 \boldsymbol{z}_\mathrm{I} \times_4 \boldsymbol{z}_\mathrm{II}$. The output of the $\tau$ element is expressed as $\sum_{\nu=1}^{\omega} \sum_{\lambda,\mu=1}^{d} w_{\tau,\nu,\lambda,\mu}^{[2,2]} b_{[1],\omega} z_{\mathrm{I},\lambda} z_{\mathrm{II},\mu}$. Similarly, the product $\hat{\boldsymbol{\mathcal{W}}}^{[2,2]} \times_2 \boldsymbol{b}_{[1]} \times_3$

$z_{II} \times_4 z_I$ has output $\sum_{\nu=1}^{\omega} \sum_{\lambda,\mu=1}^{d} w_{\tau,\nu,\mu,\lambda}^{[2,2]} b_{[1],\omega} z_{I,\lambda} z_{II,\mu}$ for the $\tau$ element. Notice that the only change in the two expressions is the permutation of the third and forth modes of the tensor; the rest of the expression remains the same. Therefore, to account for this symmetry we factorize the term $\boldsymbol{\mathcal{W}}^{[2,2]}$ as the sum of two terms and assume that each term is multiplied by the respective terms. Let $\boldsymbol{W}_{(1)}^{[2,2]} = \boldsymbol{C}\Big\{ \boldsymbol{A}_{[2,I]} \odot \Big[ \Big( \boldsymbol{A}_{[1,II]} \odot \boldsymbol{B}_{[1]} \Big) \boldsymbol{V}_{[2]} \Big] + \boldsymbol{A}_{[2,II]} \odot$

$\Big[ \Big( \boldsymbol{A}_{[1,I]} \odot \boldsymbol{B}_{[1]} \Big) \boldsymbol{V}_{[2]} \Big] \Big\}^{T}$.

The parameters $\boldsymbol{C} \in \mathbb{R}^{o \times k}, \boldsymbol{A}_{[n,\phi]} \in \mathbb{R}^{d \times k}, \boldsymbol{V}_{[n]} \in \mathbb{R}^{k \times k}, \boldsymbol{B}_{[n]} \in \mathbb{R}^{\omega \times k}$ for $n = 1, 2$ and $\phi = \{I, II\}$ are learnable. Collecting all the terms above and extracting $\boldsymbol{C}$ as a common factor (we ommit $\boldsymbol{C}$ below to avoid cluttering the notation):

$$(\boldsymbol{A}_{[2,II]} \odot \boldsymbol{B}_{[2]})^{T} (\boldsymbol{z}_{II} \odot \boldsymbol{b}_{[2]}) + (\boldsymbol{A}_{[2,I]} \odot \boldsymbol{B}_{[2]})^{T} (\boldsymbol{z}_{I} \odot \boldsymbol{b}_{[2]}) +$$

$$\Big\{ \boldsymbol{A}_{[2,II]} \odot \Big[ \Big( \boldsymbol{A}_{[1,II]} \odot \boldsymbol{B}_{[1]} \Big) \boldsymbol{V}_{[2]} \Big] \Big\}^{T} (\boldsymbol{z}_{II} \odot \boldsymbol{z}_{II} \odot \boldsymbol{b}_{[1]}) +$$

$$\Big\{ \boldsymbol{A}_{[2,I]} \odot \Big[ \Big( \boldsymbol{A}_{[1,I]} \odot \boldsymbol{B}_{[1]} \Big) \boldsymbol{V}_{[2]} \Big] \Big\}^{T} (\boldsymbol{z}_{I} \odot \boldsymbol{z}_{I} \odot \boldsymbol{b}_{[1]}) +$$

$$\Big\{ \boldsymbol{A}_{[2,I]} \odot \Big[ \Big( \boldsymbol{A}_{[1,II]} \odot \boldsymbol{B}_{[1]} \Big) \boldsymbol{V}_{[2]} \Big] \Big\}^{T} (\boldsymbol{z}_{I} \odot \boldsymbol{z}_{II} \odot \boldsymbol{b}_{[1]}) + \qquad (13)$$

$$\Big\{ \boldsymbol{A}_{[2,II]} \odot \Big[ \Big( \boldsymbol{A}_{[1,I]} \odot \boldsymbol{B}_{[1]} \Big) \boldsymbol{V}_{[2]} \Big] \Big\}^{T} (\boldsymbol{z}_{II} \odot \boldsymbol{z}_{I} \odot \boldsymbol{b}_{[1]}) =$$

$$\Big( \boldsymbol{A}_{[2,II]}^{T} \boldsymbol{z}_{II} + \boldsymbol{A}_{[2,I]}^{T} \boldsymbol{z}_{I} \Big) * \Big( \boldsymbol{B}_{[2]}^{T} \boldsymbol{b}_{[2]} \Big) +$$

$$\Big( \boldsymbol{A}_{[2,II]}^{T} \boldsymbol{z}_{II} + \boldsymbol{A}_{[2,I]}^{T} \boldsymbol{z}_{I} \Big) * \Big\{ \boldsymbol{V}_{[2]}^{T} \Big[ \Big( \boldsymbol{A}_{[1,II]}^{T} \boldsymbol{z}_{II} + \boldsymbol{A}_{[1,I]}^{T} \boldsymbol{z}_{I} \Big) * \Big( \boldsymbol{B}_{[1]}^{T} \boldsymbol{b}_{[1]} \Big) \Big] \Big\}$$

The last equation is precisely a recursive equation that can be expressed with the Fig. 5 or equivalently the generalized recursive relationship below.

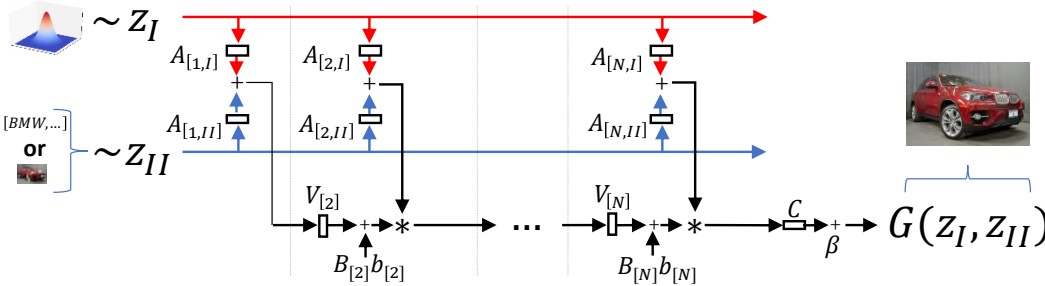

Figure 5: Abstract schematic for $N^{th}$ order approximation of $\boldsymbol{x} = G(\boldsymbol{z}_I, \boldsymbol{z}_{II})$ with Nested-MVP model. The inputs $\boldsymbol{z}_I, \boldsymbol{z}_{II}$ are symmetric in our formulation. We denote with $\boldsymbol{z}_I$ a sample from the noise distribution (e.g., Gaussian), while $\boldsymbol{z}_{II}$ symbolizes a sample from a conditional input (e.g., a class label or a low-resolution image).

**Recursive relationship**: The recursive formula for the Nested-MVP model with arbitrary expansion order $N \in \mathbb{N}$ is the following:

$$\boldsymbol{x}_n = \Big( \boldsymbol{A}_{[n,I]}^{T} \boldsymbol{z}_I + \boldsymbol{A}_{[n,II]}^{T} \boldsymbol{z}_{II} \Big) * \Big( \boldsymbol{V}_{[n]}^{T} \boldsymbol{x}_{n-1} + \boldsymbol{B}_{[n]}^{T} \boldsymbol{b}_{[n]} \Big) \qquad (14)$$

where $n \in [2, N]$ and $\boldsymbol{x}_1 = \left( \boldsymbol{A}^T_{[1,I]} \boldsymbol{z}_I + \boldsymbol{A}^T_{[1,II]} \boldsymbol{z}_{II} \right) * \left( \boldsymbol{B}^T_{[1]} \boldsymbol{b}_{[1]} \right)$. The parameters $\boldsymbol{C} \in \mathbb{R}^{o \times k}, \boldsymbol{A}_{[n,\phi]} \in \mathbb{R}^{d \times k}, \boldsymbol{V}_{[n]} \in \mathbb{R}^{k \times k}, \boldsymbol{B}_{[n]} \in \mathbb{R}^{\omega \times k}$ for $\phi = \{I, II\}$ are learnable. Then, the output $\boldsymbol{x} = \boldsymbol{C}\boldsymbol{x}_N + \boldsymbol{\beta}$.

The Nested-MVP model manifests an alternative network that relies on slightly modified assumptions on the decomposition. Thus, changing the underlying assumptions of the decomposition can modify the resulting network. This can be an important tool for domain-specific applications, e.g., when the domain-knowledge should be inserted in the last layers.

## C    BEYOND TWO VARIABLES

Frequently, more than one conditional inputs are required (Yu et al., 2018b; Xu et al., 2017; Maximov et al., 2020). In such tasks, the aforementioned framework can be generalized to more than two input variables. We demonstrate how this is possible with three variables; then it can trivially extended to an arbitrary number of input variables.

Let $\boldsymbol{z}_I, \boldsymbol{z}_{II}, \boldsymbol{z}_{III} \in \mathbb{K}^d$ denote the three input variables. We aim to learn a function that captures the higher-order interactions of the input variables. The polynomial of expansion order $N \in \mathbb{N}$ with output $\boldsymbol{x} \in \mathbb{R}^o$ has the form:

$$\boldsymbol{x} = G(\boldsymbol{z}_I, \boldsymbol{z}_{II}, \boldsymbol{z}_{III}) = \sum_{n=1}^{N} \sum_{\rho=1}^{n+1} \sum_{\delta=\rho}^{n+1} \left( \boldsymbol{\mathcal{W}}^{[n,\rho,\delta]} \prod_{j=2}^{\rho} \times_j \boldsymbol{z}_I \prod_{\tau=\rho+1}^{\delta} \times_\tau \boldsymbol{z}_{II} \prod_{\zeta=\delta+1}^{n+1} \times_\zeta \boldsymbol{z}_{III} \right) + \boldsymbol{\beta} \quad (15)$$

where $\boldsymbol{\beta} \in \mathbb{R}^o$ and $\boldsymbol{\mathcal{W}}^{[n,\rho,\delta]} \in \mathbb{R}^{o \times \prod_{m=1}^{n} \times_m d}$ (for $n \in [1, N]$ and $\rho, \delta \in [1, n+1]$) are the learnable parameters. As in the two-variable input, the unknown parameters increase exponentially. To that end, we utilize a joint factorization with factor sharing. The recursive relationship of such a factorization is:

$$\boldsymbol{x}_n = \boldsymbol{x}_{n-1} + \left( \boldsymbol{U}^T_{[n,I]} \boldsymbol{z}_I + \boldsymbol{U}^T_{[n,II]} \boldsymbol{z}_{II} + \boldsymbol{U}^T_{[n,III]} \boldsymbol{z}_{III} \right) * \boldsymbol{x}_{n-1} \quad (16)$$

for $n = 2, \ldots, N$ with $\boldsymbol{x}_1 = \boldsymbol{U}^T_{[1,I]} \boldsymbol{z}_I + \boldsymbol{U}^T_{[1,II]} \boldsymbol{z}_{II} + \boldsymbol{U}^T_{[1,III]} \boldsymbol{z}_{III}$ and $\boldsymbol{x} = \boldsymbol{C}\boldsymbol{x}_N + \boldsymbol{\beta}$.

Notice that the pattern (for each order) is similar to the two-variable input: a) a different embedding is found for each input variable, b) the embeddings are added together, c) the result is multiplied elementwise with the representation of the previous order.

## D    CONCATENATION OF INPUTS

A popular method used for conditional generation is to concatenate the conditional input with the noise labels. However, as we showcase below, concatenation has two significant drawbacks when compared to our framework. To explain those, we will define a concatenation model.

Let $\boldsymbol{z}_I \in \mathbb{K}_1^{d_1}, \boldsymbol{z}_{II} \in \mathbb{K}_2^{d_2}$ where $\mathbb{K}_1, \mathbb{K}_2$ can be a subset of real or natural numbers. The output of a concatenation layer is $x = \boldsymbol{P}^T \left[ \boldsymbol{z}_I; \boldsymbol{z}_{II} \right]^T$ where the symbol ';' denotes the concatenation and $\boldsymbol{P} \in \mathbb{R}^{(d_1 d_2) \times o}$ is an affine transformation on the concatenated vector. The $j^{th}$ output is $x_j = \sum_{\tau=1}^{d_1} p_{\tau,j} z_{I,\tau} + \sum_{\tau=1}^{d_2} p_{\tau+d_1,j} z_{II,\tau}$.

Therefore, the two differences from the concatenation case are:

- If the input variables are concatenated together we obtain an additive format, not a multiplicative that can capture cross-term correlations. That is, the multiplicative format does allow achieving higher-order auto- and cross- term correlations.

- The concatenation changes the dimensionality of the embedding space. Specifically, the input space has dimensionality $d_1 \cdot d_2$. That has a significant toll on the size of the filters (i.e., it increases the learnable parameters), while still having an additive impact. On the contrary, our framework does not change the dimensionality of the embedding spaces.

# E  IN-DEPTH DIFFERENCES FROM Π-NET

In the next few paragraphs, we conduct an in-depth analysis of the differences between Π-Net and MVP. The analysis assumes knowledge of the proposed model, i.e., (2).

Chrysos et al. (2020) introduce Π-Net as a polynomial expansion of a single input variable. Their goal is to model functions $\boldsymbol{x} = G(\boldsymbol{z})$ as high-order polynomial expansions of $\boldsymbol{z}$. Their focus is towards using a single-input variable $\boldsymbol{z}$, which can be noise in case of image generation or an image in discriminative experiments. The authors express the StyleGAN architecture (Karras et al., 2019) as a polynomial expansion, while they advocate that the impressive results can be attributed to the polynomial expansion.

To facilitate the in-depth analysis, the recursive relationship that corresponds to (2) is provided below. An $N^{th}$ order expansion in Π-Net is expressed as:

$$\boldsymbol{x}_n = \left(\boldsymbol{\Lambda}_{[n]}^T \boldsymbol{z}\right) * \boldsymbol{x}_{n-1} + \boldsymbol{x}_{n-1} \tag{17}$$

for $n = 2, \ldots, N$ with $\boldsymbol{x}_1 = \boldsymbol{\Lambda}_{[1]}^T \boldsymbol{z}$ and $\boldsymbol{x} = \boldsymbol{\Gamma} \boldsymbol{x}_N + \boldsymbol{\beta}$. The parameters $\boldsymbol{\Lambda}, \boldsymbol{\Gamma}$ are learnable.

In this work, we focus on conditional data generation, i.e., there are multiple input variables available as auxiliary information. The trivial application of Π-Net would be to concatenate all the $M$ input variables $\boldsymbol{z}_{\mathrm{I}}, \boldsymbol{z}_{\mathrm{II}}, \boldsymbol{z}_{\mathrm{III}}, \ldots$. The input variable $\boldsymbol{z}$ becomes $\boldsymbol{z} = \left[\boldsymbol{z}_{\mathrm{I}}; \boldsymbol{z}_{\mathrm{II}}; \boldsymbol{z}_{\mathrm{III}}; \ldots\right]$, where the symbol ';' denotes the concatenation. Then, the polynomial expansion of Π-Net can be learned on the concatenated $\boldsymbol{z}$. However, there are four significant reasons that we believe that this is not as flexible as the proposed MVP.

When we refer to Π-Net below, we refer to the model with concatenated input. In addition, let $\boldsymbol{z}_{\mathrm{I}} \in \mathbb{K}_1^{d_1}, \boldsymbol{z}_{\mathrm{II}} \in \mathbb{K}_2^{d_2}$ denote the input variables where $\mathbb{K}_1, \mathbb{K}_2$ can be a subset of real or natural numbers.

**Parameter sharing:**  MVP allows additional flexibility in the structure of the architecture, since MVP utilizes a different projection layer for each input variable. We utilize this flexibility to share the parameters of the conditional input variable; as we detail in (19), we set $\boldsymbol{U}_{[n,II]} = \boldsymbol{U}_{[1,II]}$ on (2). If we want to perform a similar sharing in Π-Net, the formulation equivalent to (17) would be $(\lambda_{[n]})_i = (\lambda_{[1]})_i$ for $i = d_1, \ldots, d_1 + d_2$. However, sharing only part of the matrix might be challenging. Additionally, when $\boldsymbol{\Lambda}$ is a convolution, the sharing pattern is not straightforward to be computed. Therefore, MVP enables additional flexibility to the model, which is hard to be included in Π-Net.

**Inductive bias:**  The inductive bias is crucial in machine learning (Zhao et al., 2018), however concatenating the variables restricts the flexibility of the model (i.e. Π-Net). To illustrate that, let us use the super-resolution experiments as an example. The input variable $\boldsymbol{z}_{\mathrm{I}}$ is the noise vector and $\boldsymbol{z}_{\mathrm{II}}$ is the (vectorized) low-resolution image. If we concatenate the two variables, then we should use a fully-connected (dense) layer, which does not model well the spatial correlations. Instead, with MVP, we use a fully-connected layer for the noise vector and a convolution for $\boldsymbol{z}_{\mathrm{II}}$ (low-resolution image). The convolution reduces the number of parameters and captures the spatial correlations in the image. Thus, by concatenating the variables, we reduce the flexibility of the model.

**Dimensionality of the inputs:**  The dimensionality of the inputs might vary orders of magnitude, which might create an imbalance during learning. For instance, in class-conditional generation concatenating the one-hot labels in the input does not scale well when there are hundreds of classes (Odena et al., 2017). We observe a similar phenomenon in class-conditional generation: in Cars196 (with 196 classes) the performance of Π-Net deteriorates considerably when compared to its (relative) performance in CIFAR10 (with 10 classes). On the contrary, MVP does not fuse the elements of the input variables directly, but it projects them into a subspace appropriate for adding them.

**Order of expansion with respect to each variable:**  Frequently, the two inputs do not require the same order of expansion. Without loss of generality, assume that we need correlations up to $N_I$ and $N_{II}$ order (with $N_I < N_{II}$) from $\boldsymbol{z}_{\mathrm{I}}$ and $\boldsymbol{z}_{\mathrm{II}}$ respectively. MVP includes a different transformation

for each variable, i.e., $\boldsymbol{U}_{[n,I]}$ for $\boldsymbol{z}_I$ and $\boldsymbol{U}_{[n,II]}$ for $\boldsymbol{z}_{II}$. Then, we can set $\boldsymbol{U}_{[n,I]} = 0$ for $n > N_I$. On the contrary, the concatenation of inputs (in $\Pi$-Net) constrains the expansion to have the same order with respect to each variable.

All in all, we can use concatenation to fuse variables and use $\Pi$-Net, however an inherently multivariate model is more flexible and can better encode the types of inductive bias required for conditional data generation.

## F    DIFFERENCES FROM OTHER NETWORKS CAST AS POLYNOMIAL NEURAL NETWORKS

A number of networks with impressive results have emerged in (conditional) data generation the last few years. Three such networks that are particularly interesting in our context are Karras et al. (2019); Park et al. (2019); Chen et al. (2019). We analyze below each method and how it relates to polynomial expansions:

- Karras et al. (2019) propose an Adaptive instance normalization (AdaIN) method for unsupervised image generation. An AdaIN layer expresses a second-order interaction[3]: $\boldsymbol{h} = (\boldsymbol{\Lambda}^T \boldsymbol{w}) * n(c(\boldsymbol{h}_{in}))$, where $n$ is a normalization, $c$ the convolution operator and $\boldsymbol{w}$ is the transformed noise $\boldsymbol{w} = MLP(\boldsymbol{z}_I)$ (mapping network). The parameters $\boldsymbol{\Lambda}$ are learnable, while $\boldsymbol{h}_{in}$ is the input to the AdaIN. Stacking AdaIN layers results in a polynomial expansion with a single variable.

- Chen et al. (2019) propose a normalization method, called sBN, to stabilize the GAN training. The method performs a 'self-modulation' with respect to the noise variable and optionally the conditional variable in the class-conditional generation setting. Henceforth, we focus on the class-conditional setting that is closer to our work. sBN injects the network layers with a multiplicative interaction of the input variables. Specifically, sBN projects the conditional variable into the space of the variable $\boldsymbol{z}_I$ through an embedding function. Then, the interaction of the two vector-like variables is passed through a fully-connected layer (and a ReLU activation function); the result is injected into the network through the batch normalization parameters. If cast as a polynomial expansion, a network with sBN layers expresses a single polynomial expansion[4]

- Park et al. (2019) introduce a spatially-adaptive normalization, i.e., SPADE, to improve semantic image synthesis. Their model, referred to as SPADE in the remainder of this work, assumes a semantic layout as a conditional input that facilitates the image generation. We analyze in sec. F.1 how to obtain the formulation of their spatially-adaptive normalization. If cast as a polynomial expansion, SPADE expresses a polynomial expansion with respect to the conditional variable.

The aforementioned works propose or modify the batch normalization layer to improve the performance or stabilize the training, while in our work we propose the multivariate polynomial as a general function approximation technique for conditional data generation. Nevertheless, given the interpretation of the previous works in the perspective of polynomials, we still can express them as special cases of MVP. Methodologically, there are **two significant limitations** that none of the aforementioned works tackle:

- The aforementioned architectures focus on no or one conditional variable. Extending the frameworks to **multiple conditional variables** might not be trivial, while MVP naturally extends to arbitrarily many conditional variables.

- Even though the aforementioned three architectures use (implicitly) a polynomial expansion, a significant factor is the order of the expansion. In our work, the **product of polynomials** enables capturing higher-order correlations without increasing the amount of layers substantially (sec. 3.2).

---

[3]The formulation is derived from the public implementation of the authors.

[4]In MVP, we do not learn a single embedding function for the conditional variable. In addition, we do not project the (transformed) conditional variable to the space of the noise-variable. Both of these can be achieved by making simplifying assumptions on the factor matrices of MVP.

In addition to the aforementioned methodological differences, *our work is the only polynomial expansion that conducts experiments on a variety of conditional data generation tasks*. Thus, we both demonstrate methodologically and verify experimentally that MVP can be used for a wide range of conditional data generation tasks.

### F.1 IN-DEPTH DIFFERENCES FROM SPADE

In the next few paragraphs, we conduct an in-depth analysis of the differences between SPADE and MVP.

Park et al. (2019) introduce a spatially-adaptive normalization, i.e., SPADE, to improve semantic image synthesis. Their model, referred to as SPADE in the remainder of this work, assumes a semantic layout as a conditional input that facilitates the image generation.

The $n^{th}$ model block applies a normalization on the representation $\boldsymbol{x}_{n-1}$ of the previous layer and then it performs an elementwise multiplication with a transformed semantic layout. The transformed semantic layout can be denoted as $\boldsymbol{A}_{[n,II]}^T\boldsymbol{z}_{\text{II}}$ where $\boldsymbol{z}_{\text{II}}$ denotes the conditional input to the generator. The output of this elementwise multiplication is then propagated to the next model block that performs the same operations. Stacking $N$ such blocks results in an $N^{th}$ order polynomial expansion which is expressed as:

$$\boldsymbol{x}_n = \left(\boldsymbol{A}_{[n,II]}^T\boldsymbol{z}_{\text{II}}\right) * \left(\boldsymbol{V}_{[n]}^T\boldsymbol{x}_{n-1} + \boldsymbol{B}_{[n]}^T\boldsymbol{b}_{[n]}\right) \tag{18}$$

where $n \in [2, N]$ and $\boldsymbol{x}_1 = \boldsymbol{A}_{[1,I]}^T\boldsymbol{z}_{\text{I}}$. The parameters $\boldsymbol{C} \in \mathbb{R}^{o \times k}, \boldsymbol{A}_{[n,\phi]} \in \mathbb{R}^{d \times k}, \boldsymbol{V}_{[n]} \in \mathbb{R}^{k \times k}, \boldsymbol{B}_{[n]} \in \mathbb{R}^{\omega \times k}$ for $\phi = \{I, II\}$ are learnable. Then, the output $\boldsymbol{x} = \boldsymbol{C}\boldsymbol{x}_N + \boldsymbol{\beta}$.

SPADE as expressed in (18) resembles one of the proposed models of MVP (specifically (14)). In particular, it expresses a polynomial with respect to the conditional variable. The parameters $\boldsymbol{A}_{[n,I]}$ are set as zero, which means that there are no higher-order correlations with respect to the input variable $\boldsymbol{z}_{\text{I}}$. Therefore, our work bears the following differences from Park et al. (2019):

- SPADE proposes a normalization scheme that is only applied to semantic image generation. On the contrary, our proposed MVP can be applied to any conditional data generation task, e.g., class-conditional generation or image-to-image translation.

- **SPADE is a special case of MVP**. In particular, by setting i) $\boldsymbol{A}_{[1,II]}$ equal to zero, ii) $\boldsymbol{A}_{[n,I]}$ in (14) equal to zero, we obtain SPADE. In addition, MVP allows different assumptions on the decompositions which lead to an alternative structure, such as (2).

- SPADE proposes a polynomial expansion with respect to a single variable. On the other hand, our model can extend to an arbitrary number of input variables to account for auxiliary labels, e.g., (16).

- Even though SPADE models higher-order correlations of the conditional variable, it still does not leverage the higher-order correlations of the representations (e.g., as in the product of polynomials) and hence without activation functions it might not work as well as the two-variable expansion.

Park et al. (2019) exhibit impressive generation results with large-scale computing (i.e., they report results using NVIDIA DGX with 8 V100 GPUs). Our goal is not to compete in computationally heavy, large-scale experiments, but rather to illustrate the benefits of the generic formulation of MVP.

SPADE is an important baseline for our work. In particular, we augment SPADE in wo ways: a) by extending it to accept both continuous and discrete variables in $\boldsymbol{z}_{\text{II}}$ and b) by adding polynomial terms with respect to the input variable $\boldsymbol{z}_{\text{I}}$. The latter model is referred to as SPADE-MVP (details on the next section).

## G    EXPERIMENTAL DETAILS

**Metrics:**    The two most popular metrics (Lucic et al., 2018; Creswell et al., 2018) for evaluation of the synthesized images are the Inception Score (IS) (Salimans et al., 2016) and the Frechet Inception

Distance (FID) (Heusel et al., 2017). The metrics utilize the pretrained Inception network (Szegedy et al., 2015) to extract representations of the synthesized images. FID assumes that the representations extracted follow a Gaussian distribution and matches the statistics (i.e., mean and variance) of the representations between real and synthesized samples. Alternative evaluation metrics have been reported as inaccurate, e.g., in Theis et al. (2016), thus we use the IS and FID. Following the standard practice of the literature, the IS is computed by synthesizing $5,000$ samples, while the FID is computed using $10,000$ samples.

The IS is used exclusively for images of natural scenes as a metric. The reasoning behind that is that the Inception network has been trained on images of natural scenes. On the contrary, the FID metric relies on the first and second-order moments of the representations, which are considered more robust to different types of images. Hence, we only report IS for the CIFAR10 related experiments, while for the rest the FID is reported.

**Dataset details:** There are five main datasets used in this work:

- Large-scale CelebFaces Attributes (or *CelebA* for short) (Liu et al., 2015) is a large-scale face attributes dataset with $202,000$ celebrity images. We use $160,000$ images for training our method.
- *Cars196* (Krause et al., 2013) is a dataset that includes different models of cars in different positions and backgrounds. Cars196 has $16,000$ images, while the images have substantially more variation than CelebA faces.
- *CIFAR10* (Krizhevsky et al., 2014) contains $60,000$ images of natural scenes. Each image is of resolution $32 \times 32 \times 3$ and is classified in one of the $10$ classes. CIFAR10 is frequently used as a benchmark for image generation.
- The Street View House Numbers dataset (or *SVHN* for short) (Netzer et al., 2011) has $100,000$ images of digits ($73,257$ of which for training). SVHN includes color house-number images which are classified in 10 classes; each class corresponds to a digit $0$ to $9$. SVHN images are diverse (e.g., with respect to background, scale).
- *MNIST* (LeCun et al., 1998) consists of images with handwritten digits. Each images depicts a single digit (annotated from $0$ to $9$) in a $28 \times 28$ resolution. The dataset includes $60,000$ images for training.
- *Shoes* (Yu & Grauman, 2014; Xie & Tu, 2015) consists of $50,000$ images of shoes, where the edges of each shoe are extracted (Isola et al., 2017).
- *Handbags* (Zhu et al., 2016; Xie & Tu, 2015) consists of more than $130,000$ images of handbag items. The edges have been computed for each image and used as conditional input to the generator (Isola et al., 2017).
- *Anime characters* dataset (Jin et al., 2017) consists of anime characters that are generated based on specific attributes, e.g., hair color. The public version used[5] contains annotations on the hair color and the eye color. We consider 7 classes on the hair color and 6 classes on the eye color, with a total of $14,000$ training images.

All the images of CelebA, Cars196, Shoes and Handbags are resized to $64 \times 64$ resolution.

**Architectures:** The discriminator structure is left the same for each experiment, we focus only on the generator architecture. All the architectures are based on two different generator schemes, i.e., the SNGAN (Miyato & Koyama, 2018) and the polynomial expansion of Chrysos et al. (2020) that does not include activation functions in the generator.

The variants of the generator of SNGAN are described below:

- **SNGAN** (Miyato & Koyama, 2018): The generator consists of a convolution, followed by three residual blocks. The discriminator is also based on successive residual blocks. The public implementation of SNGAN with conditional batch normalization (CBN) is used as the baseline.

---

[5]The version is downloaded following the instructions of `https://github.com/bchao1/Anime-Generation`.

- **SNGAN-MVP** [proposed]: We convert the resnet-based generator of SNGAN to an MVP model. To obtain MVP, the SNGAN is modified in two ways: a) the Conditional Batch Normalization (CBN) is converted into batch normalization (Ioffe & Szegedy, 2015), b) the injections of the two embeddings (from the inputs) are added after each residual block, i.e. the formula of (2). In other words, the generator is converted to a product of two-variable polynomials.

- **SNGAN-CONC**: Based on SNGAN-MVP, we replace each Hadamard product with a concatenation. This implements the variant mentioned in sec. D.

- **SNGAN-SPADE** (Park et al., 2019): As described in sec. F.1, SPADE is a polynomial with respect to the conditional variable $z_{II}$. The generator of SNGAN-MVP is modified to perform the Hadamard product with respect to the conditional variable every time.

The variants of the generator of $\Pi$-Net are described below:

- **$\Pi$-Net** (Chrysos et al., 2020): The generator is based on a product of polynomials. The first polynomials use fully-connected connections, while the next few polynomials use cross-correlations. The discriminator is based on the residual blocks of SNGAN. We stress out that the generator does not include any activation functions apart from a hyperbolic tangent in the output space for normalization. The authors advocate that this exhibits the expressivity of the designed model.

- **$\Pi$-Net-SICONC**: The generator structure is based on $\Pi$-Net with two modifications: a) the Conditional Batch Normalization is converted into batch normalization (Ioffe & Szegedy, 2015), b) the second-input is concatenated with the first (i.e., the noise) in the input of the generator. Thus, this is a single variable polynomial, i.e., a $\Pi$-Net, where the second-input is vectorized and concatenated with the first. This baseline implements the $\Pi$-Net described in sec. E.

- **MVP** [proposed]: The generator of $\Pi$-Net is converted to an MVP model with two modifications: a) the Conditional Batch Normalization is converted into batch normalization (Ioffe & Szegedy, 2015), b) instead of having a Hadamard product with a single variable as in $\Pi$-Net, the formula with the two-variable input (e.g., (2)) is followed.

- **GAN-CONC**: Based on MVP, each Hadamard product is replaced by a concatenation. This implements the variant mentioned in sec. D.

- **GAN-ADD**: Based on MVP, each Hadamard product is replaced by an addition. This modifies (14) to $x_n = \left( A_{[n,I]}^T z_I + A_{[n,II]}^T z_{II} \right) + \left( V_{[n]}^T x_{n-1} + B_{[n]}^T b_{[n]} \right)$.

- **SPADE** (Park et al., 2019): As described in sec. F.1, SPADE defines a polynomial with respect to the conditional variable $z_{II}$. The generator of $\Pi$-Net is modified to perform the Hadamard product with respect to the conditional variable every time.

- **SPADE-MVP** [proposed]: This is a variant we develop to bridge the gap between SPADE and the proposed MVP. Specifically, we augment the aforementioned SPADE twofold: a) the dense layers in the input space are converted into a polynomial with respect to the variable $z_I$ and b) we also convert the polynomial in the output (i.e., the rightmost polynomial in the Fig. 6 schematics) to a polynomial with respect to the variable $z_I$. This model captures higher-order correlations of the variable $z_I$ that SPADE did not not originally include. This model still includes single variable polynomials, however the input in each polynomial varies and is not only the conditional variable.

The two baselines GAN-CONC and GAN-ADD capture only additive correlations, hence they cannot effectively model complex distributions without activation functions. Nevertheless, they are added as a reference point to emphasize the benefits of higher-order polynomial expansions.

An abstract schematic of the generators that are in the form of products of polynomials is depicted in Fig. 6. Notice that the compared methods from the literature use polynomials of a single variable, while we propose a polynomial with an arbitrary number of inputs (e.g., two-input shown in the schematic).

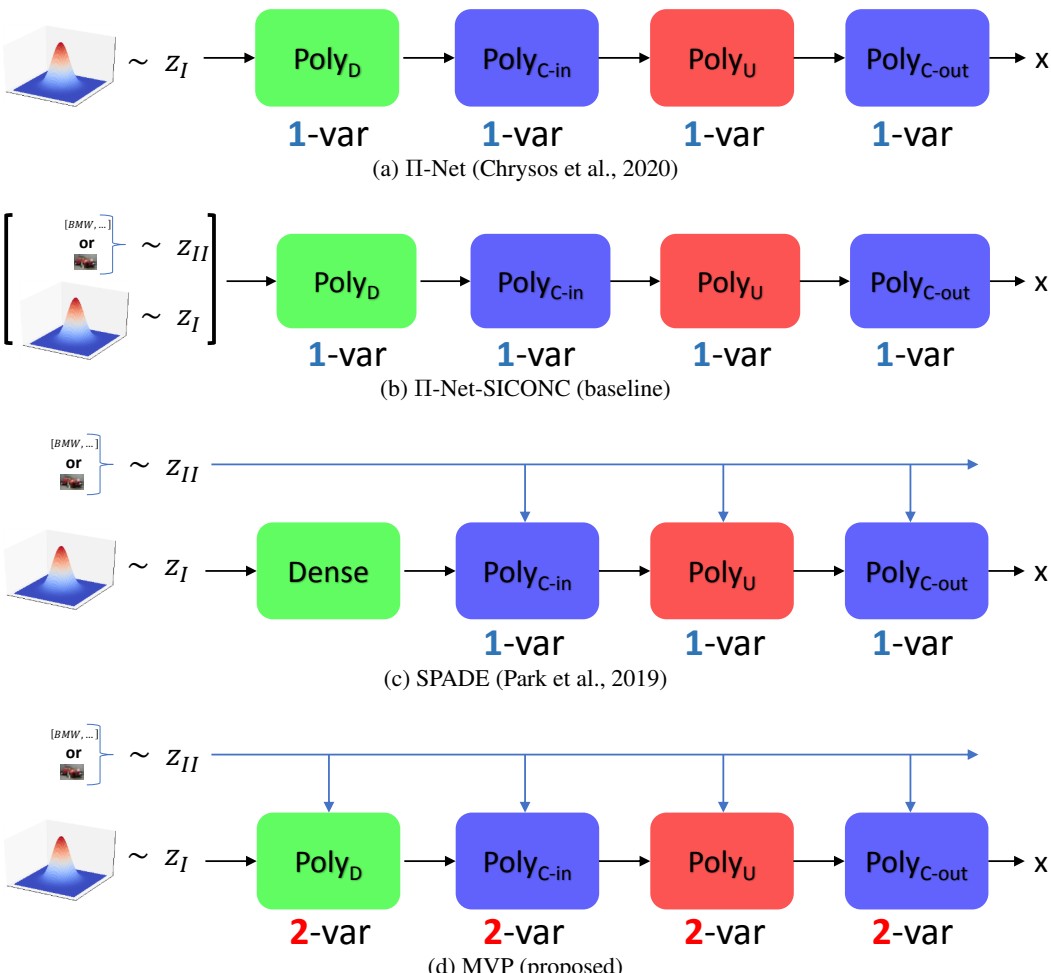

Figure 6: Abstract schematic of the different compared generators. All the generators are products of polynomials. Each colored box represents a different type of polynomial, i.e., the green box symbolizes polynomial(s) with dense layers, the blue box denotes convolutional or cross-correlation layers. The red box includes the up-sampling layers. (a) Π-Net implements a single-variable polynomial for modeling functions $x = G(z)$. Π-Net enables class-conditional generation by using conditional batch normalization (CBN). (b) An alternative to CBN is to concatenate the conditional variable in the input, as in Π-Net-SICONC. This also enables the non-discrete conditional variables (e.g., low-resolution images) to be concatenated. (c) SPADE implements a single-variable polynomial for conditional image generation. The polynomial is built with respect to the conditional variable $z_{II}$. This is substantially different from the polynomial with multiple-input variables, i.e., MVP. Two additional differences are that (i) SPADE is motivated as a spatially-adaptive method (i.e., for continuous conditional variables), while MVP can be used both for discrete and for continuous type variables, (ii) there is no polynomial in the dense layers in the SPADE. However, as illustrated in Π-Net converting the dense layers into a higher-order polynomial can further boost the performance. (d) The proposed generator structure.

**Implementation details of MVP:**   Throughout this work, we reserve the symbol $z_{II}$ for the conditional input (e.g., a class label). In each polynomial, we reduce further the parameters by using the same embedding for the conditional variables. That is expressed as:

$$\boldsymbol{U}_{[n,II]} = \boldsymbol{U}_{[1,II]} \tag{19}$$

for $n = 2, \ldots, N$. Equivalently, that would be $\boldsymbol{A}_{[n,II]} = \boldsymbol{A}_{[1,II]}$ in (14). Additionally, Nested-MVP performed better in our preliminary experiments, thus we use (14) to design each polynomial. Given

the aforementioned sharing, the $N^{th}$ order expansion is described by:

$$\boldsymbol{x}_n = \left( \boldsymbol{A}_{[n,I]}^T \boldsymbol{z}_{\mathrm{I}} + \boldsymbol{A}_{[1,II]}^T \boldsymbol{z}_{\mathrm{II}} \right) * \left( \boldsymbol{V}_{[n]}^T \boldsymbol{x}_{n-1} + \boldsymbol{B}_{[n]}^T \boldsymbol{b}_{[n]} \right) \tag{20}$$

for $n = 2, \ldots, N$. Lastly, the factor $\boldsymbol{A}_{[1,II]}$ is a convolutional layer in the case of continuous conditional input, while it is a fully-connected layer in the case of discrete conditional input.

# H  ADDITIONAL EXPERIMENTS

Additional experiments and visualizations are provided in this section. Additional visualizations for class-conditional generation are provided in sec. H.1. An additional experiment with class-conditional generation with SVHN digits is performed in sec. H.2. An experiment that learns the translation of MNIST to SVHN digits is conducted in sec. H.3. To explore further the image-to-image translation, two additional experiments are conducted in sec. H.4. An attribute-guided generation is performed in sec. H.5 to illustrate the benefit of our framework with respect to multiple, discrete conditional inputs. This is further extended in sec. H.6, where an experiment with mixed conditional input is conducted. Finally, an additional diversity-inducing regularization term is used to assess whether it can further boost the diversity the synthesized images in sec. H.7.

## H.1  ADDITIONAL VISUALIZATIONS IN CLASS-CONDITIONAL GENERATION

In Fig. 7 the qualitative results of the compared methods in class-conditional generation on CIFAR10 are shared. Both the generator of SNGAN and ours have activation functions in this experiment.

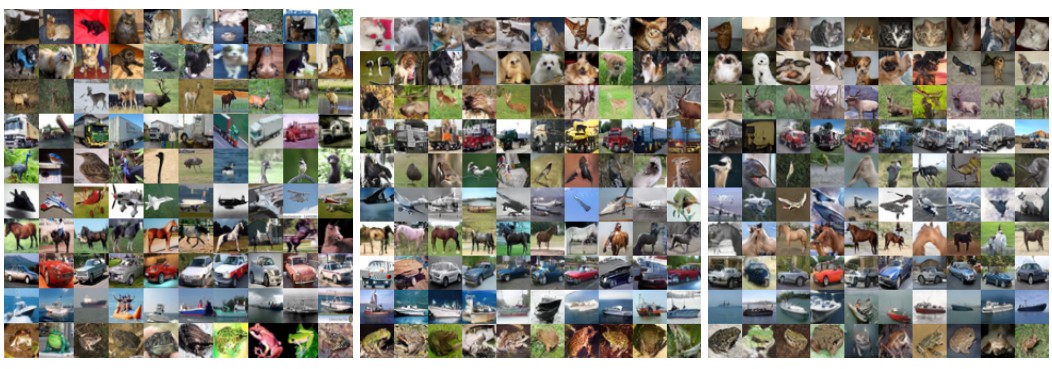

(a) Ground-truth samples          (b) SNGAN (Miyato et al., 2018)          (c) MVP

Figure 7: Qualitative results on CIFAR10. Each row depicts random samples from a single class.

In Fig. 8 samples from the baseline $\Pi$-Net (Chrysos et al., 2020) and our method are depicted for the class-conditional generation on CIFAR10. The images have a substantial difference. Similarly, in Fig. 9 a visual comparison between $\Pi$-Net and MVP is exhibited in Cars196 dataset. To our knowledge, no framework in the past has demonstrated such expressivity; MVP synthesizes images that approximate the quality of synthesized images from networks with activation functions.

In Fig. 10, an inter-class interpolation of various compared methods in CIFAR10 are visualized. The illustrations of the intermediate images in SNGAN-CONC and SNGAN-ADD are either blurry or not realistic. On the contrary, in SPADE and MVP the higher-order polynomial expansion results in more realistic intermediate images. Nevertheless, MVP results in sharper shapes and images even in the intermediate results when compared to SPADE.

## H.2  CLASS-CONDITIONAL GENERATION ON HOUSE DIGITS

An experiment on class-conditional generation with SVHN is conducted below. SVHN images include (substantial) blur or other distortions, which insert noise in the distribution to be learned. In addition, some images contain contain a central digit (i.e., based on which the class is assigned), and

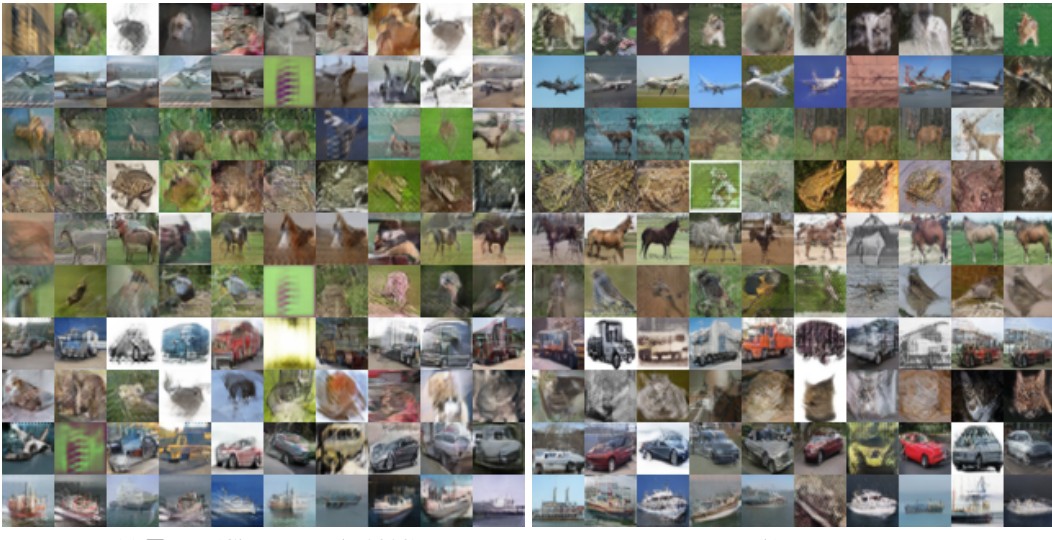

(a) Π-Net (Chrysos et al., 2020)                    (b) MVP

Figure 8: Qualitative results on CIFAR10 when the generator does not include activation functions between the layers. Each row depicts random samples from a single class; the same class is depicted in each pair of images. For instance, the last row corresponds to boats.

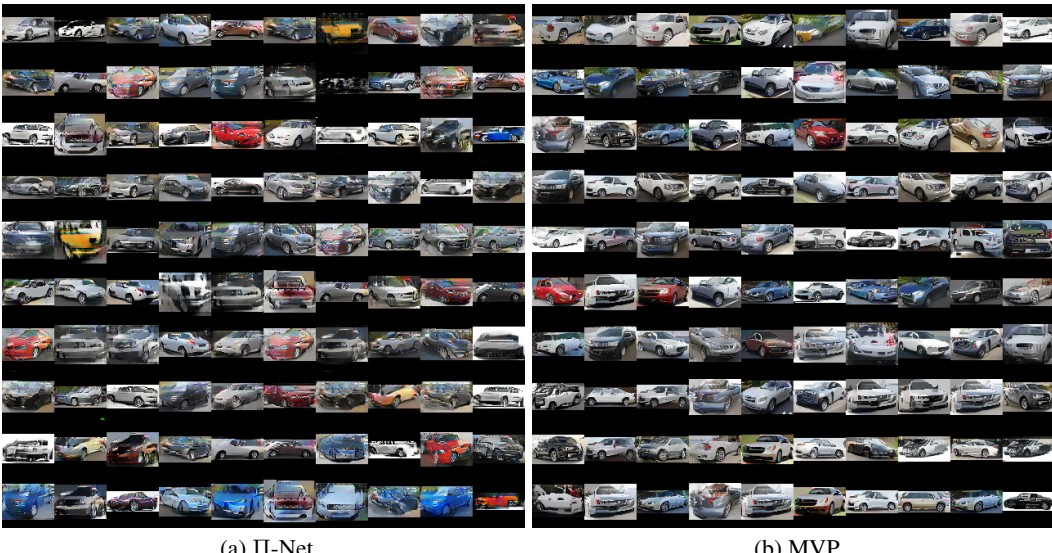

(a) Π-Net                    (b) MVP

Figure 9: Qualitative results on Cars196 when the generator does not include activation functions between the layers. Each row depicts random samples from a single class; the same class is depicted in each pair of images. The differences between the synthesized images are dramatic.

partial visibility of other digits. Therefore, the generation of digits of SVHN is challenging for a generator without activation functions between the layers.

Our framework, e.g., equation 14, does not include any activation functions. To verify the expressivity of our framework, we maintain the same setting for this experiment. Particularly, **the generator does not have activation functions between the layers**; there is only a hyperbolic tangent in the output space for normalization. The generator receives a noise sample and a class as input, i.e., it is a class-conditional polynomial generator.

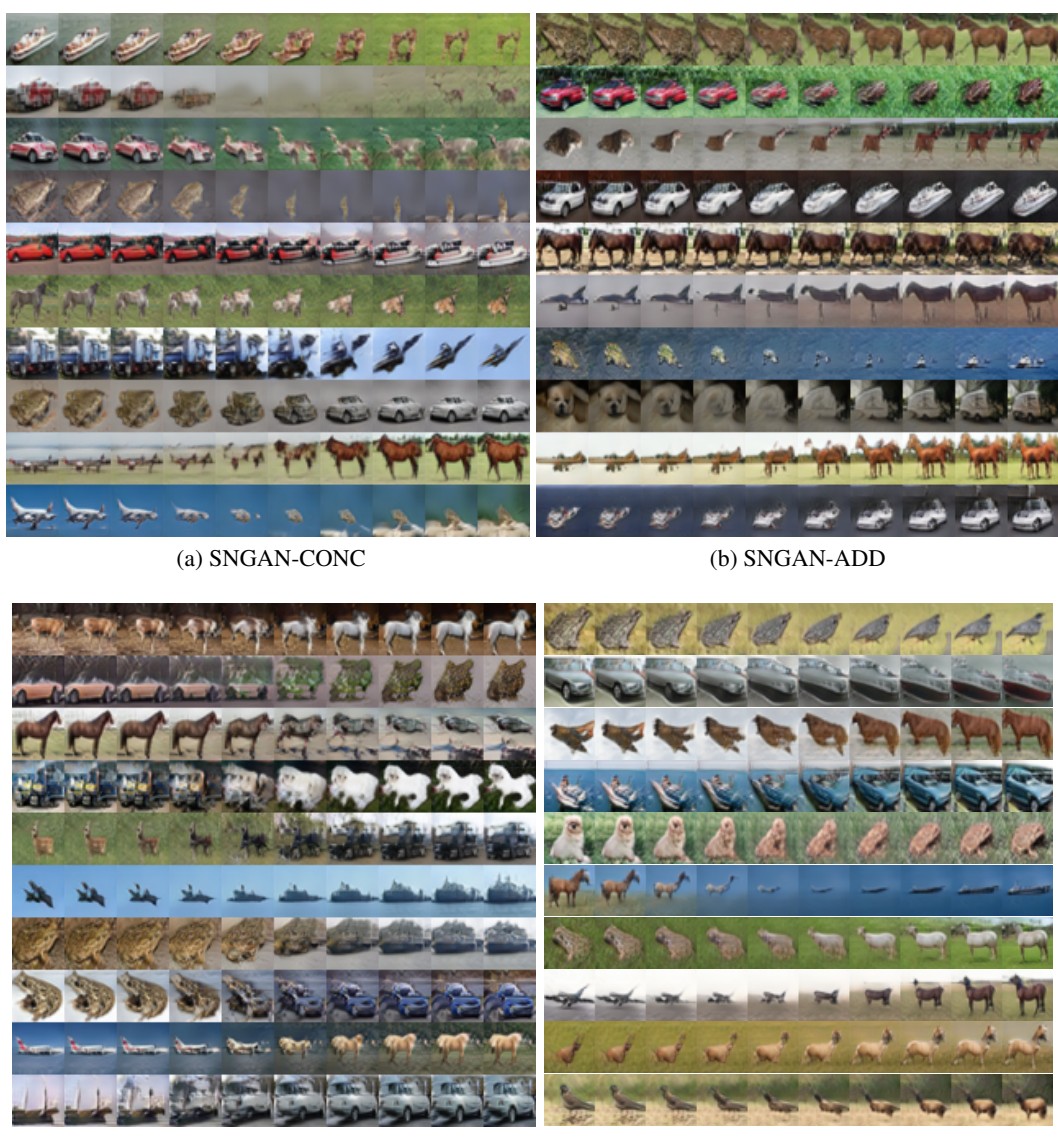

Figure 10: Inter-class linear interpolations across different methods. In inter-class interpolation, the class labels of the leftmost and rightmost images are one-hot vectors, while the rest are interpolated in-between; the resulting images are visualized. Many of the intermediate images in SNGAN-CONC and SNGAN-ADD are either blurry or not realistic. On the contrary, in SPADE and MVP the higher-order polynomial expansion results in more realistic intermediate images. Nevertheless, MVP results in sharper shapes and images even in the intermediate results when compared to SPADE.

The results in Fig. 12(b) illustrate that despite the noise, MVP learns the distribution. As mentioned in the main paper, our formulation enables both inter-class and intra-class interpolations naturally. In the inter-class interpolation the noise $z_{\mathrm{I}}$ is fixed, while the class $z_{\mathrm{II}}$ is interpolated. In Fig. 12(d) several inter-class interpolations are visualized. The visualization exhibits that our framework is able to synthesize realistic images even with inter-class interpolations.

## H.3    TRANSLATION OF MNIST DIGITS TO SVHN DIGITS

An experiment on image translation from the domain of binary digits to house numbers is conducted below. The images of MNIST are used as the source domain (i.e., the conditional variable $z_{\mathrm{II}}$), while

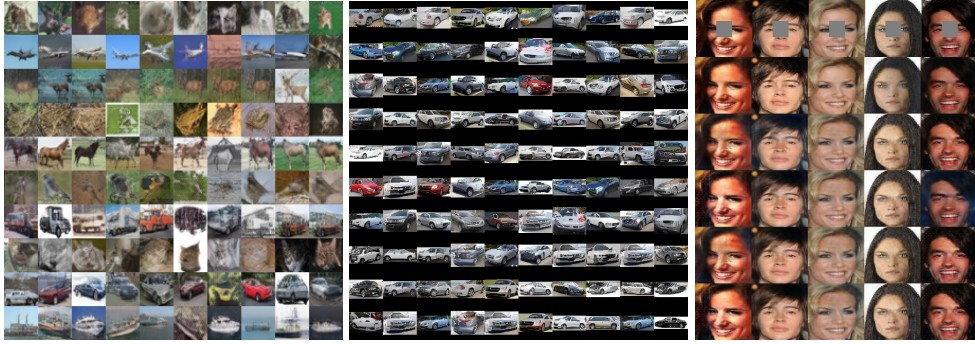

(a) Class-conditional generation      (b) Class-conditional generation      (c) Block-inpainting

Figure 11: Synthesized images by MVP in the (a), (b) class-conditional generation (sec. 4.1) and (b) block-inpainting (sec. 4.2). The networks do not include activation functions between the layers. In class-conditional generation, each row depicts a single class. Notice how the MVP synthesizes diverse images even in the absence of activation functions.

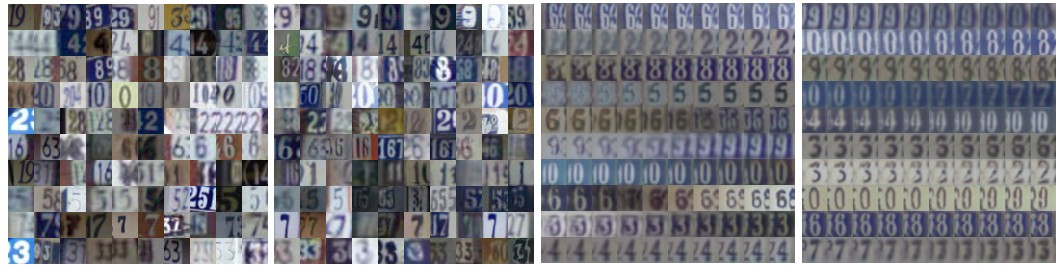

(a) Ground-truth samples      (b) MVP      (c) Intra-class interpolation (d) Inter-class interpolation

Figure 12: Synthesized images by MVP in the class-conditional SVHN: (a) Ground-truth samples, (b) Random samples where each row depicts the same class, (c) Intra-class linear interpolation from a source (leftmost image) to the target (rightmost image), (d) inter-class linear interpolation. In inter-class interpolation, the class labels of the leftmost and rightmost images are one-hot vectors, while the rest are interpolated in-between; the resulting images are visualized. In all three cases ((b)-(d)), MVP synthesizes realistic images.

the images of SVHN are used as the target domain. The correspondence of the source to the target domain is assumed to be many-to-many, i.e., each MNIST digit can synthesize multiple SVHN images. No additional loss is used, the setting of continuous conditional input from sec. 4.2 is used.

The images in Fig. 13 illustrate that MVP can translate MNIST digits into SVHN digits. Additionally, for each source digit, there is a significant variation in the synthesized images.

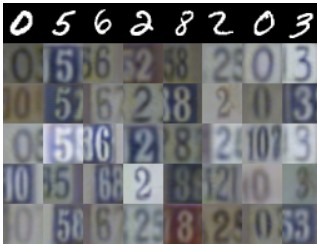

Figure 13: Qualitative results on MNIST-to-SVHN translation. The first row depicts the conditional input (i.e., a MNIST digit). The rows 2-6 depict outputs of the MVP when a noise vector is sampled per row. Notice that for each source digit, there is a significant variation in the synthesized images.

### H.4 TRANSLATION OF EDGES TO IMAGES

An additional experiment on translation is conducted, where the source domain depicts edges and the target domain is the output image. Specifically, the tasks of edges-to-handbags (on Handbags dataset) and edges-to-shoes (on Shoes dataset) have been selected Isola et al. (2017).

In this experiment, the MVP model of sec. 4.2 is utilized, i.e., a generator without activation functions between the layers. The training is conducted using *only* the adversarial loss. Visual results for both the case of edges-to-handbags and edges-to-shoes are depicted in Fig. 14. The first row depicts the conditional input $z_{\mathrm{II}}$, i.e., an edge, while the rows 2-6 depict the synthesized images. Note that in both the case of handbags and shoes there is significant variation in the synthesized images, while they follow the edges provided as input.

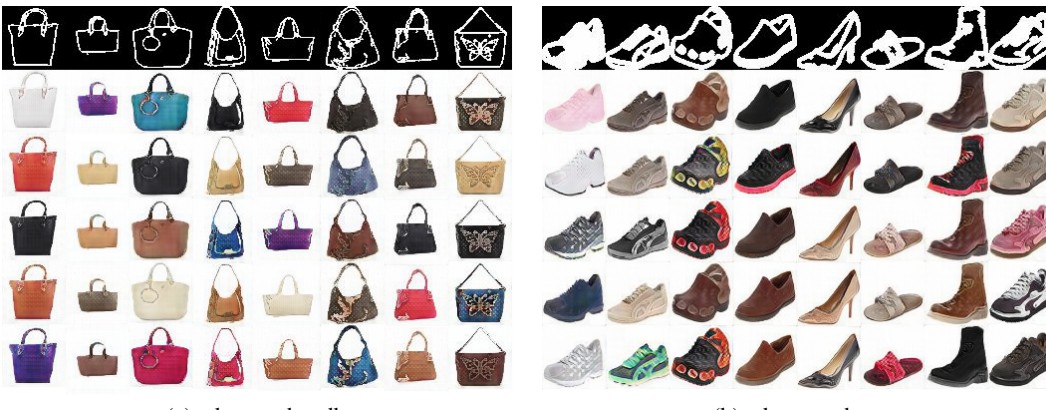

(a) edges-to-handbags          (b) edges-to-shoes

Figure 14: Qualitative results on edges-to-image translation. The first row depicts the conditional input (i.e., the edges). The rows 2-6 depict outputs of the MVP when we vary $z_I$. Notice that for each edge, there is a significant variation in the synthesized images.

### H.5 MULTIPLE, DISCRETE CONDITIONAL INPUTS

Frequently, more than one type of input conditional inputs are available. Our formulation can be extended beyond two input variables (sec. C); we experimentally verify this case. The task selected is attribute-guided generation trained on images of Anime characters. Each image is annotated with respect to the color of the eyes (6 combinations) and the color of the hair (7 combinations).

Since SPADE only accepts a single conditional variable, we should concatenate the two attributes in a single variable. We tried simply concatenating the attributes directly, but this did not work well. Instead, we can use the total number of combinations, which is the product of the individual attribute combinations, i.e., in our case the total number of combinations is 42. Obviously, this causes 'few' images to belong in each unique combination, i.e., there are 340 images on average that belong to each combination. On the contrary, there are 2380 images on average for each eye color.

SPADE and Π-Net are trained by using the two attributes in a single combination, while in our case, we consider the multiple conditional variable setting. In each case, only the generator differs depending on the compared method. In Fig. 15 few indicative images are visualized for each method; each row depicts a single combination of attributes, i.e., hair and eye color. Notice that SPADE results in a single image per combination, while in Π-Net-SINCONC there is considerable repetition in each case. The single image in SPADE can be explained by the lack of higher-order correlations with respect to the noise variable $z_I$.

In addition to the diversity of the images per combination, an image from every combination is visualized in Fig. 16. MVP synthesizes more realistic images than the compared methods of Π-Net-SINCONC and SPADE.

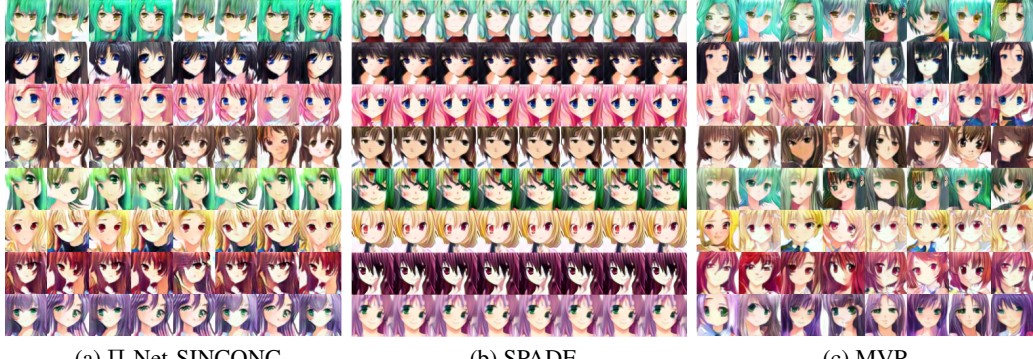

(a) Π-Net-SINCONC  (b) SPADE  (c) MVP

Figure 15: Each row depicts a single combination of attributes, i.e., hair and eye color. Please zoom-in to check the finer details. The method of SPADE synthesizes a single image per combination. Π-Net-SINCONC synthesizes few images, but not has many repeated elements, while some combinations result in unrealistic faces, e.g., the $5^{th}$ or the $7^{th}$ row. On the contrary, MVP synthesizes much more diverse images for every combination.

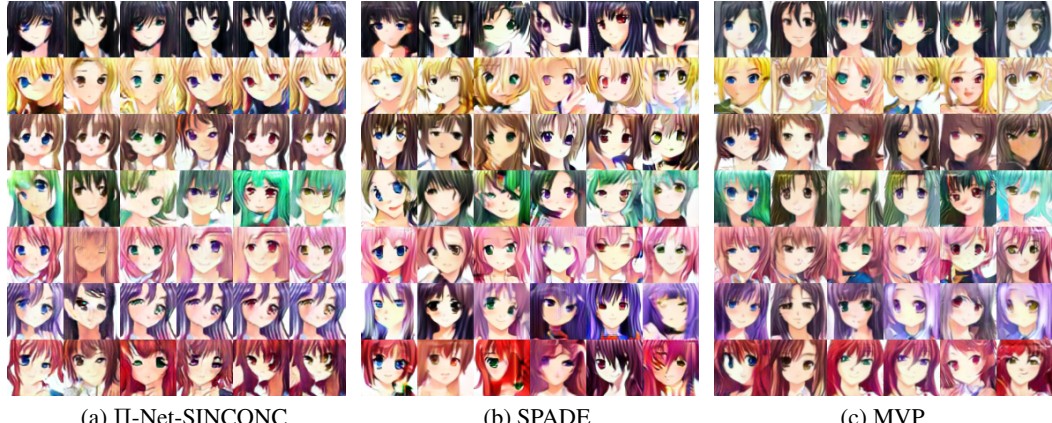

(a) Π-Net-SINCONC  (b) SPADE  (c) MVP

Figure 16: Each row depicts a single chair color, while each column depicts a single eye color. SPADE results in some combinations that do not follow the structure of the face, e.g., $3^{rd}$ column in the last row. Similarly, in Π-Net-SINCONC some of the synthesized images are not completely realistic, e.g., penultimate row. MVP synthesizes images that resemble faces for every combination.

### H.6 MULTIPLE CONDITIONAL INPUTS WITH MIXED CONDITIONAL VARIABLES

We extend the previous experiment with multiple conditional variables to the case of mixed conditional variables, i.e., there is one discrete and one continuous conditional variable. The discrete conditional variable captures the class label, while the continuous conditional variable captures the low-resolution image. Thus, the task is class-conditional super-resolution.

We use the experimental details of sec. 4.2 in super-resolution $8\times$. In Fig. 17, we visualize how for each low-resolution image the results differ depending on the randomly sampled class label. The FID in this case is 53.63, which is similar to the previous two cases. Class-conditional super-resolution (or similar tasks with multiple conditional inputs) can be of interest to the community and MVP results in high-dimensional images with large variance.

### H.7 IMPROVE DIVERSITY WITH REGULARIZATION

As emphasized in sec. I, various methods have been utilized for synthesizing more diverse images in conditional image generation tasks. A reasonable question is whether our method can be used in conjunction with such methods, since it already synthesizes diverse results. Our hypothesis is that when MVP is used in conjunction with any diversity-inducing technique, it will further improve the diversity of the synthesized images. To assess the hypothesis, we conduct an experiment on edges to images that is a popular benchmark in such diverse generation tasks (Zhu et al., 2017b; Yang et al., 2019).

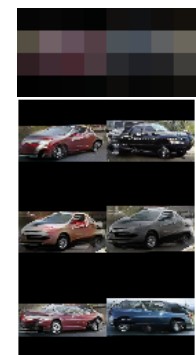

Figure 17: Three-variable input generative model.

The plug-n-play regularization term of Yang et al. (2019) is selected and added to the GAN loss during the training. The objective of the regularization term $\mathcal{L}_{reg}$ is to maximize the following term:

$$\mathcal{L}_{reg} = \min(\frac{||G(z_{\text{I},1}, z_{\text{II}}) - G(z_{\text{I},2}, z_{\text{II}})||_1}{||z_{\text{I},1} - z_{\text{I},2}||_1}, \tau) \qquad (21)$$

where $\tau$ is a predefined constant, $z_{\text{I},1}, z_{\text{I},2}$ are different noise samples. The motivation behind this term lies in encouraging the generator to produce outputs that differ when the input noise samples differ. In our experiments, we follow the implementation of the original paper with $\tau = 10$.

The regularization loss of equation 21 is added to the GAN loss; the architecture of the generator remains similar to sec. H.4. The translation task is edges-to-handbags (on Handbags dataset) and edges-to-shoes (on Shoes dataset). In Fig. 18 the synthesized images are depicted. The regularization loss causes more diverse images to be synthesized (i.e., when compared to the visualization of Fig. 14 that was trained using only the adversarial loss). For instance, in both the shoes and the handbags, new shades of blue are now synthesized, while yellow handbags can now be synthesized.

The empirical results validate the hypothesis that our model can be used in conjunction with diversity regularization losses in order to improve the results. Nevertheless, the experiment in sec. H.4 indicates that a regularization term is not necessary to synthesize images that do not ignore the noise as feed-forward generators had previously.

## I DIFFERENCE OF MVP FROM OTHER DIVERSE GENERATION TECHNIQUES

One challenge that often arises in conditional data generation is that one of the variables gets ignored by the generator (Isola et al., 2017). This has been widely acknowledged in the literature, e.g., Zhu et al. (2017b) advocates that it is hard to utilize a simple architecture, like Isola et al. (2017), with noise. A similar conclusion is drawn in InfoGAN (Chen et al., 2016) where the authors explicitly mention that additional losses are required, otherwise the generator is 'free to ignore' the additional variables. To mitigate this, a variety of methods have been developed. We summarize the most prominent methods from the literature, starting from image-to-image translation methods:

- BicycleGAN (Zhu et al., 2017b) proposes a framework that can synthesize diverse images in image-to-image translation. The framework contains 2 encoders, 1 decoder and 2 discriminators. This results in multiple loss terms (e.g., eq.9 of the paper). Interestingly, the authors utilize a separate training scheme for the encoder-decoder and the second encoder

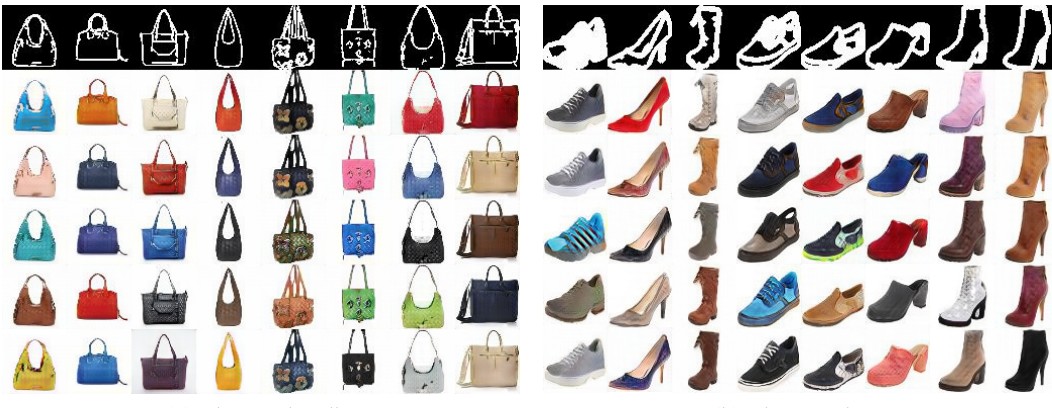

(a) edges-to-handbags                    (b) edges-to-shoes

Figure 18: Qualitative results on edges-to-image translation with regularization loss for diverse generation (sec. H.7). The first row depicts the conditional input (i.e., the edges). The rows 2-6 depict outputs of the MVP when we vary $z_I$. Diverse images are synthesized for each edge. The regularization loss results in 'new' shades of blue to emerge in the synthesized images in both the shoes and the handbags cases.

as training together 'hides the information of the latent code without learning meaningful modes'.

- Almahairi et al. (2018) augment the deterministic mapping of CycleGAN (Zhu et al., 2017a) with a marginal matching loss. The framework learns diverse mappings utilizing the additional encoders. The framework includes 4 encoders, 2 decoders and 2 discriminators.

- MUNIT (Huang et al., 2018) focuses on diverse generation in unpaired image-to-image translation. MUNIT demonstrates impressive translation results, while the inverse translation is also learnt simultaneously. That is, in case of edges-to-shoes, the translation shoes-to-edges is also learnt during the training. The mapping learnt comes at the cost of multiple network modules. Particularly, MUNIT includes 2 encoders, 2 decoders, 2 discriminators for learning. This also results in multiple loss terms (e.g., eq.5 of the paper) along with additional hyper-parameters and network parameters.

- Drit++ (Lee et al., 2020) extends unpaired image-to-image translation with disentangled representation learning, while they allow multi-domain image-to-image translations. Drit++ uses 4 encoders, 2 decoders, 2 discriminators for learning. Similarly to the previous methods, this results in multiple loss terms (e.g., eq.6-7 of the paper) and additional hyper-parameters.

- Choi et al. (2020) introduce a method that supports multiple target domains. The method includes four modules: a generator, a mapping network, a style encoder and a discriminator. All modules (apart from the generator) include domain-specific sub-networks in case of multiple target domains. To ensure diverse generation, Choi et al. (2020) utilize a regularization loss (i.e., eq. 3 of the paper), while their final objective consists of multiple loss terms.

The aforementioned frameworks contain additional network modules for training, which also results in additional hyper-parameters in the loss-function and the network architecture. Furthermore, the frameworks focus exclusively on image-to-image translation and not all conditional generation cases, e.g., they do not tackle class-conditional or attribute-based generation.

An interesting technique for diverse, class-conditional generation is the self-conditional GAN of Liu et al. (2020). The method conditions the generator with pseudo-labels that are automatically derived from clustering on the feature space of the discriminator. This enables the generator to synthesize more diverse samples. This method is orthogonal to our, i.e., the generator of Liu et al. (2020) can be replaced with MVP.

Using regularization terms in the loss function has been an alternative way to achieve diverse generation. Mao et al. (2019); Yang et al. (2019) propose simple regularization terms that can be

plugged into any architecture to encourage diverse generation. Lee et al. (2019) propose two variants of a regularization term, with the 'more stable variant' requiring additional network modules.

We emphasize that our method can be used in conjunction with many of the aforementioned techniques to obtain more diverse examples. We demonstrate that this is possible in an experiment in sec. H.7.

