# OpenReview forum: "MVP: Multivariate polynomials for conditional generation"
_ICLR.cc/2021/Conference — Reject_

### Official Review · AnonReviewer1 · 2020-10-26
**Official Blind Review #1**

**Rating:** 6
**Confidence:** 4

**Review:**

**Post rebuttal (round #3)**

Thanks to the authors' effort on the rebuttal. Despite the extensive efforts, I feel the review/rebuttal iteration is not satisfactory, possibly due to some miscommunication.

To be clear, I want to re-emphasize that significant parts of my concerns were about **misleading claims** on prior work, and comparison to them was the next step.
- For example, I just wanted to clarify that the claim "type A ignores the noise and cannot learn the stochastic mapping" is wrong. The paper could simply fix the claim instead of including a massive related work section. Maybe my review also has some responsibility: I could simply say **fix** the wrong claims, instead of indirectly delivering by pointing them out.

Also, as I explicitly mentioned the concerns A,C,D,F, the rebuttal could address them point-by-point.
- In particular, I'm not convinced that cBN/sBN is **not applicable** to continuous conditions, as sBN predicts the BN parameters from the continuous latent variable.

Despite the remaining concerns, I raised the score (from 5) to 6 as the architectures with multiplicative interactions are an important and timely topic. However, I think the paper is on the borderline, and the rebuttal and revised paper could be much stronger.

------

**Summary**

This paper extends $\Pi$-Net, deep polynomial neural networks, to a multivariate setting. The proposed network, MVP, is applied to various conditional GANs, including discrete, continuous, and mixed condition scenarios.


**Pros**

- Extending $\Pi$-Net to multivariate setting is a natural and necessary research direction.
- Good experimental results on class-conditional, image-conditional, and mixed (class + image)-conditional scenarios.


**Concerns/questions**

I. Novelty over $\Pi$-Net is not significant

- $\Pi$-Net has shown that deep polynomials can be useful for an unconditional generation. Extending it to conditional generation (using two variables) is quite straightforward. While the paper compares with other design choices (e.g., SICONC and SPADE), it is natural that MVP performs the best since it has the strongest expressive power.
- While the paper claims that "unifying discrete and continuous conditions" is a key property of MVP, standard conditional GANs can also handle those cases and are already discussed in the literature. For example, [1] considers image (face) + class (quantized age) conditions.
- While the paper claims that "network without activation function" is one of the main contributions, it was originally investigated and heavily discussed in $\Pi$-Net.

[1] Antipov et al. Face Aging With Conditional Generative Adversarial Networks. ICIP 2017.


II. Wrong claims on the (drawbacks of) prior work

The paper claims that prior conditional GANs: (Type A) "encoder network to obtain representations that are independent of the conditional variable" and (Type B) "directly concatenate the labels in the latent space" have several drawbacks. However, the claims are incorrect, as stated below:
- Type A does not ignore the noise and can learn the stochastic mapping (with proper training) [2,3]
- Type A can be successful for discrete conditions [1]
- Type B can scale beyond 10 class, especially with nonlinear mapping such as conditional BN [4,5]

The paper also claims that "inter-class interpolations in CIFAR10 have not emerged", but there is no backup for the claim. To verify this, the paper should compare the generated samples of standard GANs and MVP in Figure 2.

[2] Zhu et al. Toward Multimodal Image-to-Image Translation. NeurIPS 2017.\
[3] Huang et al. Multimodal Unsupervised Image-to-Image Translation. ECCV 2018.\
[4] Miyato & Koyama. cGANs with Projection Discriminator. ICLR 2018.\
[5] Brock et al. Large Scale GAN Training for High Fidelity Natural Image Synthesis. ICLR 2019.

III. Why polynomial should work better than standard GANs?

The paper claims that MVP performs better than standard GANs in various conditional generation setups.
- Is there any intuition that MVP should work better than standard GANs?
- How about the number of parameters or sampling speed? Is the comparison (in Table 2-4) fair in terms of complexity?

**Rating**

Due to the concerns above, I currently recommend a rating of 5.

---

> ### Author Response · Authors · 2020-11-11
> **Addressing the concerns of the reviewer; part 3/3**
>
> > The paper also claims that "inter-class interpolations in CIFAR10 have not emerged", but there is no backup for the claim. To verify this, the paper should compare the generated samples of standard GANs and MVP in Figure 2.
>
> We have not found any work in the literature on inter-class interpolations in CIFAR10, which is what we state. The baseline, i.e., SNGAN, uses conditional batch normalization (CBN), which results in non-trivial inter-class mixing as we explain below:
> Our (intuitive) understanding is that CBN creates a mixture of distributions (e.g., mixture of Gaussians), where each distribution includes high-dimensional representations. Inter-class interpolations between these different distributions that are (possibly) non-overlapping can be tricky.
>
> However, we add the visualizations recommended by the reviewer in the compared methods that use one-hot representations. The resulting figure is now Fig.10 of the updated manuscript. The visualization exhibits that methods that do not capture higher-order polynomial expansions result in blurrier intermediate images.
> ___________________________________________________________________________________________________________________________________________
>
> >  How about the number of parameters or sampling speed?
>
> We welcome the suggestion of the reviewer. We measure the sampling speed/parameters for both the SNGAN-based experiment and for the Pi-net-based experiment in class-conditional generation.
>
> The parameters (indicated in millions) along with the speed of generating images (indicated in millisecond) are reported. The speed is computed by running each generator in a single GPU with batch size 32; this is repeated 100 times and the mean is reported below.
>
> The table below compares the parameters and the speed for the SNGAN-based comparisons:
>
> |    Method            |Parameters (million) |Speed (in ms)                         |
> |---------------------|-----------------|-----------------------|
> |SNGAN        |4.3|17.57|
> |SNGAN-CONC    |5.3|21.69|
> |SNGAN-ADD    |4.8|19.52|
> |SNGAN-SPADE    |4.5|16.62|
> |SNGAN-MVP    |4.7|19.70|
>
> The number of parameters in the MVP and ADD are very similar (the Hadamard product is replaced with the addition), while the 'SNGAN-CONC' includes additional parameters to account for the increased channels due to the concatenation. Interestingly, SNGAN-SPADE is faster than SNGAN; this times improvement stems from conditional batch normalization (CBN).  SNGAN samples in every layer from CBN, while SPADE only performs a Hadamard product.
>
> The second table below measures the number of parameters and the speed in the class-conditional generation experiment of sec.4.1 on CIFAR10.
>
> |    Method            |Parameters (million) |Speed (in ms)                         |
> |---------------------|-----------------|-----------------------|
> |GAN-CONC    |5.6|18.79|
> |GAN-ADD    |4.8|17.87|
> |SPADE        |3.9|9.96|
> |Π-net-SINCONC    |4.8|12.14|
> |Π-net        |4.8|19.12|
> |MVP        |4.1|14.85|
>
> The results are similar to the ones reported for SNGAN. Notice that the Pi-net-SINCONC and Pi-net have similar number of parameters, but substantially different speed (over 50% overhead in Pi-net). The difference is that Π-net uses CBN, while Π-net-SINCONC concatenates the classes in the input. Overall, the Pi-net with CBN is the most costly in terms of speed.
> ___________________________________________________________________________________________________________________________________________
>
>
> ### **Summary**
> All in all, we have **updated the related work** to reflect the recommended changes, and we have answered the concerns of the reviewer on the contributions and the significance of a unified framework.  Furthermore, we **added the requested comparisons with respect to parameters and speed**, while we have added **new experiments on image-to-image translation** (i.e., edges-to-handbags and edges-to-shoes) in sec. H4 (Appendix); visualizations are provided in Fig.14. An additional experiment will be included until the weekend. The revised work is stronger, thus we would request the reviewer to adjust their rating or to pose any additional questions they might have.

---

> ### Author Response · Authors · 2020-11-11
> **Addressing the concerns of the reviewer; part 2/3**
>
> The second part of our responses on the concerns of the reviewer is added below:
>
>
> >  Type B can scale beyond 10 class, especially with nonlinear mapping such as conditional BN [4,5]
>
> Conditional BN was already used in our experiments in SNGAN (where the authors have used it in their experiments), and as we elaborate below it is a special case:
>   - Conditional BN (CBN) learns different statistics per class. For N classes, CBN requires parameters for N different means and variances. That both translates to a lot of newly introduced parameters and it only works for discrete classes, i.e., it cannot be done for continuous conditional variables.
>   - BigGAN [5] synthesizes photo-realistic results through large-scale training. Even though we do not claim to outperform such a work, from a technical standpoint, their technique is equivalent to our ‘GAN-CONC’ (details on sec.G of the Appendix). In short, [5] captures the additive correlations (please check sec. D In the Appendix) and not the higher-order correlations.
>
> Overall, we have already considered the CBN in our experimental comparison (class-conditional generation), however it cannot be generalized to continuous conditional variables.
> ___________________________________________________________________________________________________________________________________________
>
>
> >  Is there any intuition that MVP should work better than standard GANs?
>
> We offer two perspectives on why MVP works better than standard GANs:
>
> 1.  To our knowledge standard generators (e.g., CNN-like) rely exclusively on activation functions to capture non-linearities, while MVP can ALSO utilize the higher-order correlations that emerge in the polynomial expansion. Thus, MVP can combine the higher-order correlations with activation functions; that explains the improved results exhibited (e.g., CIFAR10 with SNGAN).
>
> 2.  Jayakumar et al [6] prove that even a second-order polynomial can expand the hypothesis space of ‘standard’ feed-forward neural networks, hence using a higher-order polynomial expansion will even further enrich the capabilities of learning complex distributions.
>
>
> The additional experimental results and validations are addressed in the next part (part 3/3).
>
>
>
> [1] Antipov et al. Face Aging With Conditional Generative Adversarial Networks. ICIP 2017.
>
> [2] Zhu et al. Toward Multimodal Image-to-Image Translation. NeurIPS 2017.
>
> [3] Huang et al. Multimodal Unsupervised Image-to-Image Translation. ECCV 2018.
>
> [4] Miyato & Koyama. cGANs with Projection Discriminator. ICLR 2018.
>
> [5] Brock et al. Large Scale GAN Training for High Fidelity Natural Image Synthesis. ICLR 2019.
>
> [6] Jayakumar et al. Multiplicative Interactions and Where to Find Them. ICLR 2020.

---

> ### Author Response · Authors · 2020-11-12
> **Addressing the concerns of the reviewer; part 1/3**
>
> We appreciate the time and the effort of the reviewer. We address all their concerns below:
>
> > Standard conditional GANs can also handle the case of 'discrete and continuous conditions' and are already discussed in the literature. For example, [1] considers image (face) + class (quantized age) conditions.
>
> We are thankful to the reviewer for the reference (discussed now in related work in the updated paper). The architecture of [1] is well-designed for the specific task, but uses auxiliary modules (e.g., encoder or 'identity preserving' latent vector optimization approach) that cannot be generalized to different tasks (e.g., class-conditional car synthesis).  In addition, the architecture of [1] does not use Gaussian noise, so the transformation is completely deterministic given one identity (face image) and one age group.
> Thus, we believe that the point of the reviewer strengthens our argument. Out of thousands of papers written annually in GANs, only a handful of them include both discrete and continuous variables. In particular, we have never noticed a 'standard' GAN that handles discrete and continuous conditional for a broad set of experiments (e.g., without additional losses or encoders).
> ___________________________________________________________________________________________________________________________________________
>
> > The "network without activation function" is one of the main contributions, it was originally investigated and heavily discussed in Π-Net.
>
> We state that in sec.4: 'Training a generator without activation functions between the layers also emerged in Π-Net (Chrysos et al., 2020).'
>
> We do not claim we are the first to conduct such experiments. Nevertheless, we are not aware of any other conditional GANs that can perform similar experiments without activation functions.
> ___________________________________________________________________________________________________________________________________________
>
> > Π-Net has shown that deep polynomials can be useful for an unconditional generation. Extending it to conditional generation (using two variables) is quite straightforward.
>
> We respectfully disagree with the reviewer. The extensive literature of conditional GANs or other areas of deep learning has demonstrated in the last few years that the formulation (e.g., how you correlate the representations) matters. Particularly, the proposed formulation has a stronger inductive bias as empirically demonstrated in a range of experiments.
> ___________________________________________________________________________________________________________________________________________
>
>  > It is natural that MVP performs the best since it has the strongest expressive power.
>
> We agree with the reviewer that MVP would be stronger than the compared methods (e.g., SICONC and SPADE), this is why we propose it. We believe this actually as a strong point in favor of our contribution. That is, in accordance with the aforementioned argument with inductive bias we believe that this work can be used as a drop-in replacement in several tasks that SPADE or SICONC/Π-nets are currently used.
> ___________________________________________________________________________________________________________________________________________
>
> > Type A does not ignore the noise and can learn the stochastic mapping (with proper training) [2,3].
>
> We are thankful for the references (included and discussed in the updated manuscript). Despite their impressive results, the refereed works further strengthen our claim that it is hard to design a model that accounts for stochastic mapping without additional modules and additional loss terms:
>
>  - [2] includes 2 encoders, 1 decoder and 2 discriminators. This results in multiple loss terms (e.g. eq.9 of the paper). Interestingly, the authors utilize a separate training scheme for the encoder-decoder and the second encoder as training together 'hides the information of the latent code without learning meaningful modes' (sec. 4 of the paper).
>  - [3] includes 2 encoders, 2 decoders, 2 discriminators for learning. This also results in multiple loss terms (e.g., eq.5 of the paper) along with additional hyper-parameters.
>
> Instead of all the additional terms, we ONLY use the GAN loss and not complicated reconstruction terms.
>
> Furthermore, [2] already advocates that it is hard to utilize a simple architecture, like pix2pix, with noise, further indicating that a simple architecture like ours is challenging to do in standard GANs.
>
>
> [1] Antipov et al. Face Aging With Conditional Generative Adversarial Networks. ICIP 2017.
>
> [2] Zhu et al. Toward Multimodal Image-to-Image Translation. NeurIPS 2017.
>
> [3] Huang et al. Multimodal Unsupervised Image-to-Image Translation. ECCV 2018.

---

> ### Comment · AnonReviewer1 · 2020-11-16
> **Response to Rebuttal Round #1**
>
> I sincerely read the other reviews and rebuttals. I appreciate the authors' efforts to address my concerns, which resolved some of them (G). However, most of my concerns are still not fully addressed:
> 1. Novelty and/or advantage of the proposed method (B,E)
> 2. Insufficient discussion and comparison with prior work (A,C,D,F)
>
> --------------------
>
> A. discrete + continuous conditions
>
> Thanks for including the discussion on missed related work. To be clear, I cited [1] as just one example that handles both discrete and continuous conditions. There is extensive literature tackling the problem, and I think [6] would be a better example (though it learns the discrete (class) and continuous (angle, width) in an unsupervised manner).
>
> [6] Chen et al. InfoGAN: Interpretable Representation Learning by Information Maximizing Generative Adversarial Nets. NeurIPS 2016.
>
> B. contribution over $\Pi$-Net
>
> $\Pi$-Net provides new insights since it firstly shows that a polynomial network without activation is expressive enough. Though the paper first shows that it is also true for conditional GANs, the contribution is less significant than the original $\Pi$-Net.
>
> I sincerely agree that the proposed method is not trivial (though one may think it is 'quite' straightforward), and the paper validates the effectiveness of the proposed method over various design choices. However, the paper could further strengthen the contribution by discussing the "advantages" of the polynomial networks, not only showing that they are "on par" with prior non-linear activation networks.
>
> C. Type A - stochastic mapping
>
> Thanks for including the discussion on missed related work. I mentioned [2,3] as just a few examples, and there is more extensive literature on diverse generation, e.g., [7,8]. In particular, [7] shows that a simple regularizer significantly improves the diversity (though the experiments are conducted upon BicycleGAN). The paper could further strengthen the claim by demonstrating its simplicity and effectiveness over the prior methods, perhaps with the quantitative results.
>
> [7] Yang et al. Diversity-Sensitive Conditional Generative Adversarial Networks. ICLR 2019.\
> [8] Lee et al. Harmonizing Maximum Likelihood with GANs for Multimodal Conditional Generation. ICLR 2019.
>
> D. Type B - scalability
>
> The point of my initial review was clarifying the potential misleading statement "however, the model does not scale well beyond 10 classes," not comparing with the proposed method. Nevertheless, I think the point that the cBNs cannot handle continuous conditions is interesting.
>
> E. advantages over standard GANs
>
> 1. As the authors mentioned, feed-forward + non-linear activation may not be sufficient to learn high-order correlations. However, recent techniques such as cBN or SPADE (and sBN [9]) address this issue by skip connection. Namely, it is not a unique property of polynomial networks.
> 2. Jayakumar et al. [10] proved that multiplicative interactions are more expressive than sequential networks. However, it supports that prior multiplicative architectures (e.g., StyleGAN, SPADE, cBN) are effective. Polynomial networks are indeed a good example of multiplicative interactions (as also stated in [10]), but it needs more clues to convince the effectiveness over the other multiplicative architectures.
>
> [9] Chen et al. On Self Modulation for Generative Adversarial Networks. ICLR 2019.\
> [10] Jayakumar et al. Multiplicative Interactions and Where to Find Them. ICLR 2020.
>
> F. inter-class interpolation
>
> Though SNGAN-projection [11] only demonstrated the category morphing results on ImageNet, one can expect that the results could be generalized to CIFAR-10. Though no prior work explicitly presents the experiments, the claim "inter-class interpolations in CIFAR10 have not emerged" is inappropriate since no one verified it is "not emerged."
>
> Furthermore, SNGAN-projection mixes the parameters of cBNs to interpolate over the classes, as stated on page 7 of [11].
>
> [11] Miyato & Koyama. cGANs with Projection Discriminator. ICLR 2018.
>
> G. complexity of the methods
>
> Thanks for including the results. The computational complexity seems to be comparable.

---

> > ### Author Response · Authors · 2020-11-17
> > **Rebuttal round #2: Responses to Reviewer 1; part 3/3**
> >
> > > I sincerely agree that the proposed method is not trivial (though one may think it is 'quite' straightforward), and the paper validates the effectiveness of the proposed method over various design choices.
> >
> > We are thankful to the reviewer for recognizing our work.
> > ______
> >
> > ### Summary
> >
> > All in all, we appreciate the effort of the reviewer to scrutinize the differences from the related work. Our answers above detail why we believe that Π-nets (or works like SPADE, StyleGAN) do not cover our goal. Similarly, we address the questions on how our work differs from standard GANs. We would be happy to answer any follow up questions from the Reviewer. We appreciate their effort to engage with our replies and recognize our work.
> > ______
> > [6] Chen et al. InfoGAN: Interpretable Representation Learning by Information Maximizing Generative Adversarial Nets. NeurIPS 2016.
> >
> > [7] Yang et al. Diversity-Sensitive Conditional Generative Adversarial Networks. ICLR 2019.
> >
> > [8] Lee et al. Harmonizing Maximum Likelihood with GANs for Multimodal Conditional Generation. ICLR 2019.
> >
> > [9] Chen et al. On Self Modulation for Generative Adversarial Networks. ICLR 2019.
> >
> > [10] Jayakumar et al. Multiplicative Interactions and Where to Find Them. ICLR 2020.
> >
> > [11] Miyato & Koyama. cGANs with Projection Discriminator. ICLR 2018.
> >
> > [12] Karras et al. A style-based generator architecture for generative adversarial networks. CVPR 2019.
> >
> > [13]  Park et al. Semantic image synthesis with spatially-adaptive normalization. CVPR 2019.
> >
> > [14] Hanin and Rolnick. How to start training: The effect of initialization and architecture. NeuriPS 2018.
> >
> > [15] Zaeemzadeh, et al. Norm-Preservation: Why Residual Networks Can Become Extremely Deep?. T-PAMI 2020.
> >
> > [16] Balduzzi et al. The shattered gradients problem: If resnets are the answer, then what is the question?. ICML 2017.
> >
> > [17] Hardt and Ma. Identity matters in deep learning. ICLR 2017.

---

> > ### Author Response · Authors · 2020-11-17
> > **Rebuttal round #2: Responses to Reviewer 1; part 2/3**
> >
> > > The paper could further strengthen the contribution by discussing the "advantages" of the polynomial networks, not only showing that they are "on par" with prior non-linear activation networks.
> >
> > We would like to stress out that MVP is not on par with the respective non-linear networks. On the contrary, *MVP is the only alternative that generalizes to diverse tasks and does not ignore the noise* without the complicated training schemes (e.g., additional regularization losses and/or network modules) or the alternatives.
> >
> > If the reviewer refers to general arguments in favor of polynomial networks as a function approximation method, we can iterate on the arguments of Π-nets, and the theoretical proof of [10]. Our understanding is that additional work will be conducted in the future on elucidating those benefits, as a lot of architectures demonstrate state-of-the-art behavior in various tasks (e.g., Π-nets, SPADE, StyleGAN). Nevertheless, we believe that the contribution of MVP is already clear as iterated a few lines above.
> > ______
> >
> >
> > ### Novelty with respect to other conditional GANs
> >
> > > There is extensive literature tackling the discrete+continuous condition, and I think [6] would be a better example (though it learns the discrete (class) and continuous (angle, width) in an unsupervised manner).
> >
> > We are thankful for the valuable references of the reviewer. We recognize that there are some efforts in the direction we mention, and Infogan is a significant paper for disentanglement.
> >
> > However, with respect to our task, the referenced paper strengthens our argument. *Quoting from the **Infogan paper: ‘in standard GAN, the generator is free to ignore the additional latent code c** by finding a solution satisfying PG(x|c) =PG(x).’ (sec.4)*. The authors in InfoGAN utilize additional losses to regularize their method, hence we believe that this further strengthens our point. Standard GANs seem to ignore the latent code c, unless there are additional loss terms and/or additional network modules.
> >
> > As evidenced by previous replies, *several GANs have been making the observation that the standard GAN learns to ignore the second variable (depending on the task this might be the noise or the conditional variable) in the conditional case, unless additional modules and/or losses are included*. We believe this is **ample evidence to support our method** that does not require new modules or losses to learn to generate images that do not ignore the noise.
> > ______
> >
> > > Methods such as [7], [8] can generate diverse images (though the experiments in [7] are conducted upon BicycleGAN). The paper could further strengthen the claim by demonstrating its simplicity and effectiveness over the prior methods, perhaps with the quantitative results.
> >
> > The experimental comparisons already include alternative techniques that are straightforward alternatives to the proposed method. We can also try to include an additional experiment with [7]. We will update the results when the results of the experiments are available (each experiment requires several hours to run, and we have a limited amount of GPUs available).
> >
> > Nevertheless, those methods are different from ours and can be used in conjunction with [7] or [8]. In particular, both [7] and [8] are regularization terms that are added in the total loss function; in fact, one of the models in [8] (which the authors mention as the `more stable variant’) also includes additional network modules.
> >
> > These indicative methods on diverse generation have been included in sec.4.2 that we present our results on conditional generation with continuous conditional variables.
> > ______
> > > The point of my initial review was clarifying the potential misleading statement "however, the model does not scale well beyond 10 classes".
> >
> > We have noticed that there is not so much literature on class-conditional generation where they concatenate the labels. We believe it should be possible, but it would probably require additional loss term or regularization. If the reviewer has any references on the topic, we would be happy to discuss them.
> > ______
> > > Though SNGAN-projection [11] only demonstrated the category morphing results on ImageNet, one can expect that the results could be generalized to CIFAR-10. Though no prior work explicitly presents the experiments, the claim "inter-class interpolations in CIFAR10 have not emerged" is inappropriate since no one verified it is "not emerged."
> >
> > The claim has been removed from the updated manuscript, since the inter-class interpolations are visualized for many compared methods.
> > ______

---

> > ### Author Response · Authors · 2020-11-17
> > **Rebuttal round #2: Responses to Reviewer 1; part 1/3**
> >
> > ###  Novelty with respect to other polynomial networks
> >
> > > As the authors mentioned, feed-forward + non-linear activation may not be sufficient to learn high-order correlations. However, recent techniques such as cBN or SPADE (and sBN [9]) address this issue by skip connection. Namely, it is not a unique property of polynomial networks.
> >
> > We agree that polynomial networks might not be the ONLY way to capture the complex interactions required. However, the existing literature suggests that skip connections are more related to the optimization [14-17]. In that sense, we see the contribution of identity mapping (i.e., skip connections) as complementary to the one of polynomial neural networks.
> > ______
> > > Polynomial networks are indeed a good example of multiplicative interactions, but it needs more clues to convince the effectiveness over the other multiplicative architectures (e.g., StyleGAN, SPADE, sBN [9]).
> >
> > Let us summarize the contributions in each case first:
> >
> > -   Karras et al. [12] propose an Adaptive instance normalization (AdaIN) method for unsupervised image generation. As suggested in Π-nets, AdaIN captures higher-order correlations. Thus, StyleGAN is a particular form of a single-variable polynomial with a particular architecture.
> >
> > -   Park et al [13] introduce a method for spatially-adaptive normalization (SPADE) that is used for semantic image synthesis. SPADE performs conditional generation by capturing the higher-order interactions of the conditional variable. That is, it does NOT capture the higher-order interactions of the first variable.
> >
> > -   Chen et al [9] propose a normalization method (sBN) to stabilize the GAN training. The method performs a `self-modulation’ with respect to the input variable(s). sBN captures multiplicative interactions, but as the authors claim they ‘focus primarily on the unsupervised generation’. Their conditional model, which is only applied in a class-conditional setting, also assumes that the conditional variable is projected into the space of the noise-variable, while in our case we make no such assumption.
> >
> > The aforementioned works propose or modify the batch normalization layer to improve the performance or stabilize the training, while in our work we propose *the multivariate polynomial as a general function approximation technique for conditional data generation*.  The three aforementioned works do not demonstrate their relationship to polynomial expansions, while *we demonstrate that explicitly*. Namely, *we illustrate how to derive the architecture based on the coupled decomposition* (we even utilize two decompositions that result in different architectures). That means, that any practitioner could easily extend MVP or build their own new architectures. We advocate that this is a significant benefit of our work.
> >
> > Nevertheless, given the interpretation of the previous works in the perspective of polynomials, **we still can express them as special cases of MVP**. Furthermore, there are **two significant limitations that none of the aforementioned works tackle**:
> >
> > -   None of the aforementioned architectures generalizes to **more than two variables**; MVP naturally extends to arbitrarily many.
> >
> > -   None of the aforementioned works mentions or even models the **product of polynomials** which was significant in our work (discussed in the method section). That provides higher-order correlations without increasing the amount of layers substantially.
> > ______
> >
> > > Π-Net provides new insights since it firstly shows that a polynomial network without activation is expressive enough. Though the paper first shows that it is also true for conditional GANs, the contribution is less significant than the original Π-Net.
> >
> > We agree that the topic of learning without activation functions has emerged before, e.g., in Π-Nets. However, the experiments without activation functions were not our main claim.
> >
> > We argue that our work differs substantially from Π-Nets (but nevertheless we cite the paper). The **motivation and the task in our work are unrelated to the focus of Π-Nets**. Our motivation lies in proposing a conditional generator that unifies the handling of discrete and continuous conditional variables. In addition, the task is conditional data generation, i.e., from its nature it has two-variable (or multi-variable) inputs. We strongly believe that **conditional data generation is a very significant task on its own right**. Characteristically, the related work includes over 40 different methods on conditional data generation and this is not an exhaustive list as the reviewers recognize. For instance, the inverse tasks that we mention are studied for decades; MVP brings a way to learn such tasks without additional losses and we believe this is valuable. Last but not least, we demonstrate in a diverse set of tasks that MVP outperforms Π-Nets.
> > ______

---

### Official Review · AnonReviewer2 · 2020-10-26
**Clear and polished but more evidence is needed to justify the importance of MVP**

**Rating:** 5
**Confidence:** 3

**Review:**

This paper proposes a conditional generation framework (cGAN) that bridges the gap between discrete and continuous variable used in the generation. They do so by proposing a new network architecture that implements higher order multi variate polynomials (MVP). They show that MVP generalizes well to different types of conditional variables and has good expressivity even in the absence of activation functions.

Pros:
The figures are succinct and informative, giving a clear picture of the points the authors want to illustrate. I appreciate the multiple evaluations with mean and standard deviation reported.

The methods section is well-written and the intuition provided to readers who are not familiar with the concept of polynomial networks is very useful.

Cons/Comments:
Section 2.1 Discrete conditional variable: the authors claim that conditional normalization might be an obstacle towards generalizing to unseen classes. They do not show their method is able to generalize.

Section 2.1 Continuous conditional variable: The authors do not cite or mentioned papers that do multi modal image generation [1,2,3] that display diversity and do not require pixel-wise loss such as l1 or perceptual. They single out one-to-one translation models that are not designed for such tasks.

Table 2: Why is there a huge disparity between IS and FID scores of SNGAN-CONC compared to SNGAN? The IS is slightly better but the FID is significantly worse. Furthermore, the FID of the original SNGAN paper [4] is 21.7 and the FID of BigGAN [5] is 14.73. What causes the disparity the results between the papers? In my experience, BigGAN performs much better than SNGAN yet the FID of SNGAN quoted in this paper is 14.7, virtually the same as BigGAN.

Section 4.1 Resnet-based generator: The authors claim that inter-class interpolations have been done for other datasets but not CIFAR. Is BigGAN capable of doing inter-class interpolations for CIFAR since it is able to do for ImageNet?

The experiments include conditional generation based on discrete variables and continuous variables. One of the main comparisons is with SPADE. However, SPADE was designed for image generation from semantic maps, a task not tested in this paper. Super resolution was chosen as the one of the two tasks for continuous variable. This is a task not traditionally done with multi-modal output in mind. In my opinion, this is not the most useful task to base the experiments on. A more interesting task could be image-to-image translation or semantic image synthesis.

The experiments are all done at 64x64 resolution and the datasets are relatively simple compared to those used in SOTA models (512x512 ImageNet on BigGAN and 1024x1024 FFHQ on StyleGAN [6]). The quantitative improvements on the most commonly used ResNet architecture is minimal. While MVP remains expressive without activations and improves significantly using a Π net architecture, those to my knowledge are not settings widely used now in literature.


In general, this paper provides a clear unified framework for conditional image generation. The method is well explained and illustrated. However, their improvements on a commonly used architecture (ResNet) is minimal. It is my opinion that more justification and evidence is needed on why MVP is needed under common settings.

[1] Choi, Yunjey, et al. "Stargan v2: Diverse image synthesis for multiple domains." Proceedings of the IEEE/CVF Conference on Computer Vision and Pattern Recognition. 2020.

[2] Huang, Xun, et al. "Multimodal unsupervised image-to-image translation." Proceedings of the European Conference on Computer Vision (ECCV). 2018.

[3] Lee, Hsin-Ying, et al. "Drit++: Diverse image-to-image translation via disentangled representations." International Journal of Computer Vision (2020): 1-16.

[4] Miyato, Takeru, et al. "Spectral normalization for generative adversarial networks." arXiv preprint arXiv:1802.05957 (2018).

[5] Brock, Andrew, Jeff Donahue, and Karen Simonyan. "Large scale gan training for high fidelity natural image synthesis." arXiv preprint arXiv:1809.11096 (2018).

[6] Karras, Tero, Samuli Laine, and Timo Aila. "A style-based generator architecture for generative adversarial networks." Proceedings of the IEEE conference on computer vision and pattern recognition. 2019.

---

> ### Author Response · Authors · 2020-11-11
> **Addressing the concerns of the reviewer R2; part 3/3**
>
> > While MVP remains expressive without activations and improves significantly using a Π-net architecture, those to my knowledge are not settings widely used now in literature.
>
> This is the setting used by Π-nets, hence we utilize that to provide a fair comparison. In addition, conducting theoretical work in the presence of activation functions is challenging [8]. Hence, we argue that the experiments demonstrate consistently the benefits of using MVP.
> ___________________________________________________________________________________________________________________________________________
>
> > More evidence is needed to justify the importance of MVP.
>
> We appreciate the comments of the reviewer; we have addressed all their comments. In short, we summarize that our (stated) goal is not to provide state-of-the-art results, which is consistent with the policy of ICLR (please see above). Instead, we provide consistent results in a broad range of tasks. To our knowledge, the aforementioned tasks have not been considered before for generation from a single model. In addition, we augment the results based on the recommendation of the reviewer, i.e. we conduct experiments on image-to-image translation (please see the updated image-to-image results above).
>
> Our work has become significantly stronger with the revised experimental results (and related work), thus we request the reviewers to reconsider their rating.
>
>
>
>
>
>
> [1] Choi, Yunjey, et al. "Stargan v2: Diverse image synthesis for multiple domains." Proceedings of the IEEE/CVF Conference on Computer Vision and Pattern Recognition. 2020.
>
> [2] Huang, Xun, et al. "Multimodal unsupervised image-to-image translation." Proceedings of the European Conference on Computer Vision (ECCV). 2018.
>
> [3] Lee, Hsin-Ying, et al. "Drit++: Diverse image-to-image translation via disentangled representations." International Journal of Computer Vision (2020): 1-16.
>
> [4] Miyato, Takeru, et al. "Spectral normalization for generative adversarial networks." arXiv preprint arXiv:1802.05957 (2018).
>
> [5] Brock, Andrew, Jeff Donahue, and Karen Simonyan. "Large scale gan training for high fidelity natural image synthesis." arXiv preprint arXiv:1809.11096 (2018).
>
> [6] Karras, Tero, Samuli Laine, and Timo Aila. "A style-based generator architecture for generative adversarial networks." Proceedings of the IEEE conference on computer vision and pattern recognition. 2019.
>
> [7] Bousmalis, Konstantinos, et al. "Domain Separation Networks." NeurIPS 2016.
>
> [8] Arora, Sanjeev, et al. "A Convergence Analysis of Gradient Descent for Deep Linear Neural Networks." ICLR 2019.

---

> ### Author Response · Authors · 2020-11-11
> **Addressing the concerns of the reviewer R2; part 2/3**
>
>
> > Section 4.1 Resnet-based generator: The authors claim that inter-class interpolations have been done for other datasets but not CIFAR. Is BigGAN capable of doing inter-class interpolations for CIFAR since it is able to do for ImageNet?
>
> BigGAN uses conditional batch normalization (CBN), which includes a mean and a variance vector for each class (discrete choices). Our (intuitive) understanding is that this creates a mixture of distributions (e.g., mixture of Gaussians), where each distribution includes high-dimensional representations. Inter-class interpolations between these different distributions that are (possibly) non-overlapping can be tricky. In addition, neither the paper nor the source code include any further information. Hence, we are not sure that BigGAN is successful in interpolating between classes in CIFAR10.
> __________________________________________________________________________________________________________________________________
>
>
> > A more interesting task for continuous variable could be image-to-image translation or semantic image synthesis.
>
> In the revised paper, we include a number of experiments on image translation. First of all, we include an experiment of translating MNIST to SVHN digits. This is a significant baseline in domain adaptation, e.g., in [7]. The experiment is included in sec. H3 (Appendix in page 25) and we demonstrate that without any additional loss MVP can translate one MNIST digit to an SVHN equivalent.
>
> We also conduct an experiment on the more classic edges-to-image case, i.e., the conditional input includes the edges. The experiments with edges-to-handbags and edges-to-shoes are described in sec. H4 (Appendix, page 26-27).
>
> In contrast to standard practices in the literature that include additional losses or modules, e.g., in pix2pix, we only include the GAN loss for the aforementioned experiments. Yet, MVP synthesizes images that are diverse.
>
> We also share an additional video to demonstrate how we can interpolate z_I (i.e., noise) when a given shape is provided. The video is upload in this [link](https://anonymous.4open.science/r/b527f99e-5fe8-4377-8b87-615688fb656f/) and in each frame we sample one z_I, while z_II is depicted in the first row.
> ____________________________________________________________________________________________________________________________________
>
>
> > One of the main comparisons is with SPADE. SPADE was designed for image generation from semantic maps, a task not tested in this paper.
>
> In this work, we attempt to explain the success of SPADE in semantic generation. We attribute that to SPADE including higher-order correlations of the conditional input. Thus, SPADE is a generic module and consider it as such in our experiments for various tasks.
>
> As for the experiments, we have opted for classic 'inverse tasks', such as super-resolution. Those tasks are significant and have been studied for decades with both academic and commercial applications. We are not aware of any theoretical work that illustrates that semantic image generation is more challenging than the classic inverse tasks.
> ___________________________________________________________________________________________________________________________________
>
>
> > The experiments are all done at 64x64 resolution and the datasets are relatively simple compared to those used in SOTA models (512x512 ImageNet on BigGAN and 1024x1024 FFHQ on StyleGAN [6]). The quantitative improvements on the most commonly used ResNet architecture is minimal.
>
> We actually view this as a strength of our paper. Our claim is that the architectures previously proposed  cannot work well for both continuous and discrete conditional variables. We demonstrate that in a number of experiments with 'relatively simple' datasets. Conducting large-scale experiments that are computationally intensive (e.g., scaling up from 64x64 to 512x512) will not change the fact that previously proposed architectures do not work well. On the contrary, we demonstrate that our architecture can **consistently** synthesize images in both continuous and discrete conditional variables.
> __________________________________________________________________________________________________________________________________
>
>
> > The quantitative improvements on the most commonly used ResNet architecture is minimal.
>
> Obtaining state-of-the-art performance is not the goal of our paper. Nevertheless, we demonstrate that with minimal changes in the generator, we improve upon its results.
>
> This is consistent with the policy of ICLR on state-of-the-art results ('a lack of state-of-the-art results does not by itself constitute grounds for rejection. Submissions bring value to the ICLR community when they convincingly demonstrate new, relevant, impactful knowledge.': [link](https://iclr.cc/Conferences/2021/ReviewerGuide)). The reviewer recognizes that we propose a 'well explained and illustrated' unified framework, which was our goal.

---

> ### Author Response · Authors · 2020-11-11
> **Addressing the concerns of the reviewer R2; part 1/3**
>
> We appreciate the thorough review of R2. Their concerns are addressed below:
>
> > The authors claim that conditional normalization might be an obstacle towards generalizing to unseen classes.
>
> Even though the statement was correct, we have rephrased the expression to better reflect the drawbacks with respect to the comparisons we conduct in the paper. We are thankful for the remark from the reviewer.
> ___________________________________________________________________________________________________________________________________________
>
> > The authors do not mention papers that do multi modal image generation [1,2,3] that display diversity and do not require pixel-wise loss such as l1 or perceptual. They single out one-to-one translation models that are not designed for such tasks.
>
> We appreciate the references (included and discussed in the updated manuscript). The refereed works focus on (unpaired) image-to-image translation, hence they can learn very powerful mappings. However, learning such powerful mappings comes at the cost of having multiple additional modules for training along with additional loss terms (and the corresponding hyper-parameters):
>
>   - [1] includes multiple encoders and decoders for learning, which results in multiple loss terms (e.g., eq.5 of the paper).
>  - [2] includes 2 encoders, 2 decoders, 2 discriminators for learning. This also results in multiple loss terms (e.g., eq.5 of the paper) along with additional hyper-parameters.
>  - [3] uses 4 encoders, 2 decoders, 2 discriminators for learning. Similarly to the previous methods, this results in multiple loss terms (e.g., eq.6-7 of the paper) and additional hyper-parameters.
>
> Instead of all the additional terms, we ONLY use the GAN loss with one generator (decoder) and one discriminator.
> ___________________________________________________________________________________________________________________________________________
>
>
> > Why is there a huge disparity between IS and FID scores of SNGAN-CONC compared to SNGAN?
>
> We believe that this can be attributed to the difference between IS and FID. FID can measure mode collapse, while IS  reports whether synthesized images 'look' realistic.
>
> SNGAN-CONC concatenates the representation instead of using Hadamard products. That is, if each of the input representations has dimensions HxWxC (H is the height, W the width and C the channels), then the output is HxWx(2C). That means that the next layer should have 2C channels. Hence, the SNGAN-CONC has a significantly bigger capacity to 'memorize' patterns on the training data. IS only reports whether those images 'look' realistic, which can be achieved with patterns from the training data. On the other hand, the mean/variance of the FID are computed in a withheld set, so such patterns might be less effective. This is our interpretation, since evaluation in the generative models is challenging.
> ___________________________________________________________________________________________________________________________________________
>
> > The FID of the original SNGAN paper [4] is 21.7.
>
> We report the FID score (i.e., 14.7) that we obtain by using the open source code of the authors, i.e., [link](https://github.com/pfnet-research/sngan_projection). This is improved from the score the original paper reports (the lower the FID the better). In our opinion, this is because we use the conditional batch normalization for class-conditional training (different from their unsupervised CIFAR10 experiment).
> ___________________________________________________________________________________________________________________________________________
>
> > The FID of BigGAN [5] is 14.73. What causes the disparity in the results between the papers? In my experience, BigGAN performs much better than SNGAN yet the FID of SNGAN quoted in this paper is 14.7, virtually the same as BigGAN.
>
> The authors of BigGAN do not provide many details on the experiment, so we assume that it is based on class-conditional generation (like the rest of the Imagenet results). The [Github repo of the authors](https://github.com/ajbrock/BigGAN-PyTorch) is claimed to be 'unofficial', while scores with that code are not provided (also no trained models). Thus, we cannot verify the results, but even the results stated in BigGAN [5] are marginally bigger than the mean FID score of SNGAN we report.
>
> We argue that BigGAN focuses on large-scale training and not optimizing for CIFAR10, that is why their results improve SNGAN on large-scale experiments.

---

### Official Review · AnonReviewer4 · 2020-10-27
**Inappropriate experiments**

**Rating:** 5
**Confidence:** 3

**Review:**

Summary: This paper proposes a framework for generating conditional data using multivariate polynomials, which treats both discrete and continuous conditional variables in a unified way. From my perspective,  neither the effect nor the actual use is very clear.

Below are my major concerns:

1. The author compares the proposed method with multiple methods on the class-conditional generation and image2image translation tasks, but none of the comparison methods are state-of-the-art. The choice of dataset and task is not very appropriate.
Why compare with SPADE on the class conditional generation and super-resolution tasks? SPADE is the SOTA of  **semantic image synthesis** in 2019. It is unfair to compare specific methods for inappropriate tasks.
In addition, the selected dataset is also weird. Why not use a standard super-resolution dataset or use downsampled FFHQ and CelebAHQ images for face super-resolution tasks.
2. The author mentioned that the disadvantage of the traditional encoder/decoder architecture (pix2pix Isola et al., 2017) is that it ignores noise,  but in fact, this problem was solved to a certain extent as early as 2017 e.g. (BicycleGAN zhu al., 2017). In fact, in Figure 4, I did not see enough of the variability caused by the noise claimed in this paper, even though the task is so simple.
3. Although the method proposed in this paper is novel, I am not sure it really makes sense in actual complex input scenarios. Besides, the results mentioned in this paper do not show any superiority over traditional architectural design.

---

> ### Author Response · Authors · 2020-11-13
> **Addressing the concerns of the reviewer R4; part 2/2**
>
> > In fact, in Figure 4, I did not see enough of the variability caused by the noise claimed in this paper, even though the task is so simple.
>
> To better demonstrate our point, the revised paper includes a number of experiments on image translation. First of all, we include an experiment of translating MNIST to SVHN digits. This is a significant baseline in domain adaptation, e.g., in [4]. The experiment is included in sec. H3 (Appendix in page 25) and we demonstrate that without any additional loss MVP can translate one MNIST digit to an SVHN equivalent.
>
> We also conduct an experiment on the more classic edges-to-image case, i.e., the conditional input includes the edges. The experiments with edges-to-handbags and edges-to-shoes are described in sec. H4 (Appendix, page 26-27).
>
> In contrast to standard practices in the literature that include additional losses or modules, e.g., in pix2pix, we only include the GAN loss for the aforementioned experiments. Yet, MVP synthesizes images that are diverse (e.g., Fig.14).
>
> We also share an additional video to demonstrate how we can interpolate z_I (i.e., noise) when a given shape is provided. The video is upload in this [link](https://anonymous.4open.science/r/b527f99e-5fe8-4377-8b87-615688fb656f/) and in each frame we sample one z_I, while z_II is depicted in the first row.
> ________________________________________________
>
> > Although the method proposed in this paper is novel, I am not sure it really makes sense in actual complex input scenarios. Besides, the results mentioned in this paper do not show any superiority over traditional architectural design.
>
> First of all, we are thankful to the reviewer for acknowledging that our paper is novel. Once again, we repeat that according to the [reviewer guidelines](https://iclr.cc/Conferences/2021/ReviewerGuide), the proposed method does not need to have sota results to add value to the ICLR community.
>
> In addition, we would gladly cite and discuss any model that can synthesize images when trained in these diverse tasks. That is actually our point, i.e., previously separate models were built for different categories of the tasks.
>
> To sum up, we believe the tasks selected are appropriate and have significant applications, while we demonstrate consistently that the proposed model can synthesize images in a broad range of tasks. We would gladly respond to any further questions raised by the reviewer.
>
>  ________________________________________________
>
>
> [1] Nasrollahi, Kamal, and Thomas B. Moeslund. "Super-resolution: a comprehensive survey."  Machine vision and applications, 2014.
>
> [2] Wang, Zhihao, Jian Chen, and Steven CH Hoi. "Deep learning for image super-resolution: A survey." IEEE Transactions on Pattern Analysis and Machine Intelligence, 2020.
>
> [3] Anwar, Saeed, Salman Khan, and Nick Barnes. "A Deep Journey into Super-resolution: A Survey." ACM Computing Surveys (CSUR), 2020.
>
> [4] Bousmalis, Konstantinos, et al. "Domain Separation Networks." NeurIPS 2016.

---

> ### Author Response · Authors · 2020-11-13
> **Addressing the concerns of the reviewer R4; part 1/2**
>
> We appreciate the time and effort of the reviewer in understanding our work. We address their concerns below:
>
> > The author compares the proposed method with multiple methods on the class-conditional generation and image2image translation tasks.
>
> We are thankful to the reviewer for recognizing that our work conducts experiments on multiple methods.
> ________________________________________________
>
> > None of the comparison methods are state-of-the-art.
>
> Obtaining state-of-the-art performance is not the goal of our paper.
> This is consistent with the policy of ICLR on state-of-the-art results ('a lack of state-of-the-art results does not by itself constitute grounds for rejection. Submissions bring value to the ICLR community when they convincingly demonstrate new, relevant, impactful knowledge.': [link](https://iclr.cc/Conferences/2021/ReviewerGuide)). The reviewer recognizes that we propose a  novel method, which was our goal.
> ________________________________________________
>
> > Why compare with SPADE on the class conditional generation and super-resolution tasks? SPADE is the SOTA of **semantic image synthesis** in 2019.
>
> SPADE synthesizes impressive results; it is an impactful paper. In this work, we do not attempt to outperform it, but rather we attempt to explain the success of SPADE in semantic generation. We attribute the success of SPADE to the higher-order correlations of the conditional input. Thus, SPADE is a generic module and we consider it as such in our experiments for various tasks. We do NOT make any claim that our method outperforms the sota results of SPADE in semantic image generation.
>
> Given the link of SPADE to multivariate polynomials, we do consider it being relevant to our work and thus compare with it. The new perspective allows future work to extend the large-scale experiments SPADE using two-variable polynomials.
> ________________________________________________
>
> > The choice of dataset and task is not very appropriate.
>
> We respectfully disagree with the reviewer.  The 'inverse tasks', such as super-resolution or block inpainting, are classic tasks in the image processing literature.
>
> Indicatively, super-resolution has been used in a wide range of applications, such as satellite and aerial imaging, medical image processing, compressed image enhancement, fingerprint image enhancement. Several research papers, PhD theses and even books have been written about the topic. The survey of Nasrollahi and Moeslund [1] provides a comprehensive introduction to the topic, while more recent surveys [2, 3] cover the width of the latest works.
>
> However, if the reviewer has any theoretical arguments why semantic image generation is a more `appropriate’ task, we would be glad to discuss them.
> ________________________________________________
>
> > In addition, the selected dataset is also weird. Why not use a standard super-resolution dataset or use downsampled FFHQ and CelebAHQ images for face super-resolution tasks.
>
> One of the datasets selected for the super-resolution task is Celeb-A, which includes precisely faces of images. Celeb-A is one of the most highly cited papers the last few years with over 3,000 references. Celeb-A is considered the default large-scale benchmark in a wide range of machine learning papers.
> ________________________________________________
>
> > The author mentioned that the disadvantage of the traditional encoder/decoder architecture (pix2pix Isola et al., 2017) is that it ignores noise, but in fact, this problem was solved to a certain extent as early as 2017 e.g. (BicycleGAN zhu al., 2017).
>
> We have updated the related work to discuss BicycleGAN. Even though BicycleGAN synthesizes diverse images, it also includes 2 encoders, 1 decoder and 2 discriminators. This results in multiple loss terms (e.g., eq.9 of the paper). Interestingly, the authors utilize a separate training scheme for the encoder-decoder and the second encoder as training together 'hides the information of the latent code without learning meaningful modes' (sec. 4 of the paper).
>
> Furthermore, BicycleGAN already advocates that it is hard to utilize a simple architecture, like pix2pix, with noise, further indicating that a simple architecture like ours is challenging to do in standard GANs.
>
> Thus, the point of the reviewer strengthens our argument, i.e., it is hard to design a standard GAN that does not ignore the noise without additional modules or losses.

---

### Author Response · Authors · 2020-11-17
**Summary of changes in revision 1**

We appreciate the work and the feedback received so far by the reviewers. We summarize the changes/updates in the manuscript after the thorough reviews we received:

  - We have augmented the related work to include indicative unpaired image-to-image translation works and other works proposed on diverse image generation.
  -  We conduct additional experiments on image-to-image translation with edges-to-handbags and edges-to-shoes tasks. The related images are now in the manuscript, and an additional video has been shared.
  -  Additional visualization has been created to illustrate the inter-class interpolation in the compared methods.
  - The sampling speed of the compared methods has been computed.

We would be happy to address any further questions of the reviewers.

---

> ### Author Response · Authors · 2020-11-24
> **Summary of changes in revision 2**
>
> The additional feedback of Reviewer 1 has led to a significant improvement of the original submission. A number of extensions have been included, while new experiments have been conducted. Specifically:
>
> - We include a paragraph for diverse generation in the related work. We also added a table to demonstrate the differences from existing works. We emphasize that our goal is not diverse generation per se, we just observe that the proposed generator does not ignore the noise.
> - An additional experiment is conducted on edges-to-image translation, where we augment the GAN loss with the regularization term proposed in [1] (sec. H.7). That verifies our assumption that our work is complementary to the diverse generation literature.
> - We include Table 2 that illuminates the differences of polynomial-like networks with our work.
> - Sec.F (Appendix) presents the differences of our work with networks that can be cast as polynomial neural networks.
> - A new experiment on attribute-guided generation is added in sec.H.5. The dataset of Anime characters is annotated with respect to the hair color and the eye color. A comparison with both SPADE and Π-net highlights the benefits of our method.
>
> [1] Yang et al. Diversity-Sensitive Conditional Generative Adversarial Networks. ICLR 2019.

---

### Author Response · Authors · 2020-11-25
**Summary of the updated version**

We appreciate the time and feedback of the reviewers; we conducted lengthy discussion based on the (original) responses. To better summarize our contributions we can state the following:

1.  Our method provides a “novel” framework (R4); we believe this is the first work for multivariate polynomials for conditional data generation with deep neural networks.

2.  We demonstrate empirically the performance of the method, i.e., MVP, in **five tasks** (that in most cases appear in dedicated papers for each task): a) *class-conditional generation* (3 datasets), b) *super-resolution* (2 datasets), c) *block-inpainting*, d) *image-to-image translation* (3 datasets; requested by R2), e) *attribute-guided generation*. In total, there are **eight different datasets**; some of them are used in various tasks. The proposed method exhibits *consistent* improvements in every comparison with other methods.

3.  The timings (requested by R1) demonstrate that our framework does not have a significant overhead when compared to the alternatives.

4.  Reviewers pointed out some concerns with respect to the related work. The related work has since been expanded to include additional works; Table 1 and Table 2 include a concise summary of the differences with techniques for diverse generation and other polynomial-like networks.

We are sincerely grateful to the reviewers for their comments, especially with respect to the related work. We firmly believe that the updates have made our paper more valuable. We hope that the reviewers and the ACs will find the updated manuscript useful for the community.

---

### Decision · Program_Chairs · 2021-01-07
**Final Decision**

**Decision:**

Reject

**Comment:**

This paper proposes a new network architecture that implements higher order multivariate polynomials (MVP). They show that MVP generalizes well to different types of conditional variables, and can be applied to a broad range of tasks.   However, unifying discrete and continuous conditions and network without activation function are both well studied in literatures. The inappropriate discussions on the prior works and the advantages of the proposed method over prior approaches are not clearly justified.  Although outperforming SOTA is not necessary, the compared methods need to be well chosen which can provide convincing evidence on why MVP is needed under common settings.